# WHEN MORE IS LESS: UNDERSTANDING CHAIN-OF-THOUGHT LENGTH IN LLMS

**Yuyang Wu**[1,8*] **Yifei Wang**[2*] **Ziyu Ye**[3] **Tianqi Du**[1,4] **Stefanie Jegelka**[6,5] **Yisen Wang**[1,4,7†]

[1] Peking University    [2] Amazon AGI SF Lab[‡]    [3] University of Chicago
[4] State Key Lab of General Artificial Intelligence,
   School of Intelligence Science and Technology, Peking University
[5] MIT CSAIL    [6] CIT, MCML, MDSI, TU Munich
[7] Institute for Artificial Intelligence, Peking University
[8] School of EECS, Peking University

## ABSTRACT

Large Language Models (LLMs) increasingly rely on Chain-of-Thought (CoT) reasoning to solve complex problems. Contrary to the common belief that longer CoTs always improve performance, we demonstrate that **longer is not always better**. Across both real-world LLMs and theoretical models, task accuracy follows an inverted U-shaped curve with respect to CoT length: performance rises initially but declines once reasoning chains become too long. Through controlled experiments, we uncover **scaling behaviors of the optimal CoT length**: it increases with task difficulty but decreases with model capability. This exposes a significant mismatch with current practice, where supervised training often reuses the same CoT data across models and tasks without adaptivity. We further show that Reinforcement Learning (RL) can mitigate this gap by dynamically calibrating CoT length, thereby improving accuracy and offering a new perspective on differences between supervised fine-tuning and RL training. To explain these phenomena, we introduce an error-accumulation analysis that characterizes how reasoning errors propagate across steps and derives the scaling behaviors of CoT length observed empirically. Building on these insights, we show that training with optimally sized CoTs and applying length-aware filtering during inference yields substantial improvements in performance. Taken together, these findings establish a principled explanation of the "overthinking" effect and yield practical guidelines for calibrating CoT length in accordance with task complexity and model capability. Code is at https://github.com/PKU-ML/CoT-Length.

## 1 INTRODUCTION

Large language models (LLMs) have demonstrated impressive capabilities in solving complex reasoning tasks (Brown et al., 2020; Touvron et al., 2023). A central technique enabling these advances is Chain-of-Thought (CoT) reasoning (Wei et al., 2022), where models generate explicit intermediate steps to decompose complex problems into simpler, more manageable sub-problems, akin to a divide-and-conquer strategy (Zhang et al., 2024).

A widely held intuition, supported by prior studies (Fu et al., 2023; Jin et al., 2024), is that longer and more detailed CoTs generally yield better performance, especially on difficult tasks. At the same time, recent evidence shows that concise CoTs can sometimes be more effective, though often with trade-offs on challenging problems (Nayab et al., 2024). This raises a fundamental question: *does reasoning performance consistently improve as CoTs grow longer, or is there an inherent limit?*

In this paper, we provide a comprehensive answer through evidence from real-world LLMs, synthetic experiments, and theoretical modeling. We show that for CoT reasoning, *longer is not always better*. As illustrated in Figures 1a and 1b, task accuracy typically follows an *inverted U-shaped* curve with

---

[*]Equal Contribution.
[†]Corresponding Author: Yisen Wang (yisen.wang@pku.edu.cn).
[‡]This work was completed at MIT prior to Yifei Wang joining Amazon.

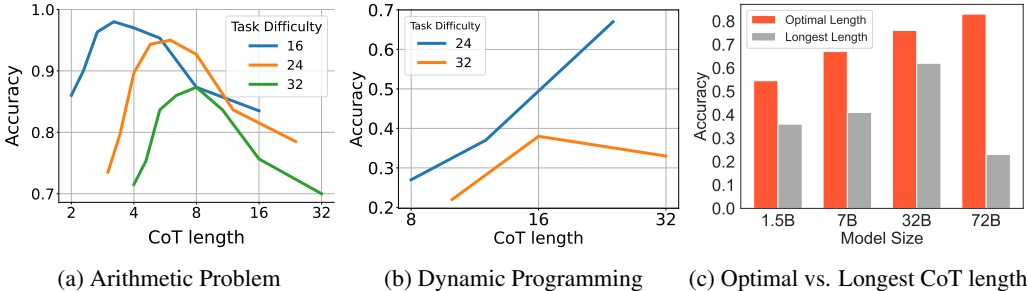

(a) Arithmetic Problem  (b) Dynamic Programming  (c) Optimal vs. Longest CoT length

Figure 1: (a) Accuracy of a 6-layer GPT-2 on arithmetic tasks, showing inverted U-shaped curves with peaks shifting to longer CoTs as task difficulty increases. (b) Accuracy of a 5-layer GPT-2 on dynamic programming tasks, also following an inverted U-curve with respect to CoT length. (c) On the MMLU STEM dataset, CoTs of optimal length significantly outperform the longest CoTs.

respect to CoT length: performance improves when the chain appropriately decomposes the task, but deteriorates when the chain becomes excessively long (due to error accumulation) or too short (leaving individual steps overly complex). This reveals the existence of an **optimal CoT length** that balances these competing forces. Identifying and calibrating to this optimal length is crucial for building reasoning models that are both efficient and accurate.

To uncover the mechanisms underlying this optimality, we design controlled experiments on arithmetic and dynamic programming tasks, and identify clear scaling behaviors: **(1)** harder tasks generally require longer CoTs to reach peak performance, **(2)** more capable models often achieve their maximum accuracy with shorter CoTs, and **(3)** solving harder tasks at the optimal length involves tackling increasingly difficult sub-tasks, motivating the need for adaptive reasoning strategies such as iterative or looping processes. Experiments on LLMs ranging from 1.5B to 72B parameters further confirm these trends. Together, these findings demonstrate that the optimal CoT length should adapt to both the problem and the model. As shown in Figure 1c, reasoning with the optimal length can significantly outperform the longest-possible CoTs (by more than 60% on a 72B model). In contrast, current practice often applies uniform CoT strategies across tasks and models during supervised learning, leading to systematic misspecification and suboptimal reasoning—sometimes causing larger models to perform worse than smaller ones. We further show that reinforcement learning (RL) can mitigate this gap by adaptively calibrating CoTs to their optimal lengths in pursuit of higher rewards. This sheds new light on why RL fine-tuning often yields superior reasoning performance and generalization compared to supervised learning (Huan et al., 2025).

To deepen our understanding, we develop a simple theoretical model based on an error-accumulation perspective: each model has a per-step success probability, so excessively long CoTs suffer from compounding errors, while overly short CoTs struggle with high per-step difficulty. This analysis not only explains the existence of an optimal length but also derives scaling laws that align closely with empirical observations. Extensions to nonlinear and stochastic error functions show the robustness of this perspective. At last, building on these insights, we demonstrate practical benefits: (i) training with optimally sized CoTs allows small models to outperform much larger ones trained on uniform-length data, and (ii) at inference time, filtering CoTs by estimated entropy yields consistent gains, improving majority-vote performance on LLaMA3-8B-Instruct and Qwen2.5-7B-Instruct.

In summary, our work makes the following contributions:

- **Longer is not always better.** We demonstrate the existence of an optimal CoT length across both real-world LLMs and synthetic tasks, challenging the prevailing intuition that performance monotonically improves with longer reasoning chains.

- **Scaling of optimal CoT length.** Through carefully controlled experiments, we systematically investigate how the optimal length depends on task difficulty and model capability, revealing consistent scaling laws: harder tasks require longer chains, while more capable models peak with shorter ones.

- **RL improves reasoning by calibrating CoT length.** We show that reinforcement learning adaptively steers CoT generation toward the optimal length, thereby explaining its superior reasoning performance compared to supervised finetuning.

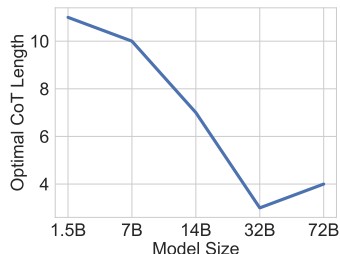 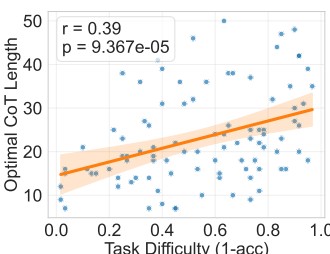 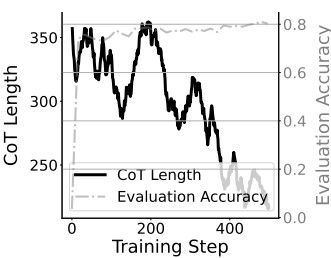

(a) Optimal CoT length vs. Model size (Qwen2.5 series).

(b) Optimal CoT length vs. Task difficulty (with the 7B model).

(c) Evolution of LLMs' CoT lengths during RL

Figure 2: Real-world observations of CoT length. (a) Larger models reach peak performance with shorter CoTs. (b) Harder tasks (lower baseline accuracy) require longer optimal CoTs, with a significant positive correlation ($p \ll 0.05$). (c) During RL training with GRPO on LeetCode-2K using Qwen2.5-7B-Instruct, average CoT length decreases as accuracy improves, suggesting that RL can also promote more efficient and concise reasoning paths.

- **An error-accumulation analysis.** We develop a simple yet useful theoretical model for understanding the observed CoT behaviors. From an error-accumulative perspective, this analysis explains the inverted U-shaped performance curve, derives the existence of an optimal CoT length, and recovers the observed scaling laws.

- **Practical implications.** We demonstrate actionable applications of our findings: (i) training with optimally sized CoT data enables smaller models to outperform larger ones trained with uniform-length chains, and (ii) a length-aware majority voting strategy that filters by entropy yields consistent gains at inference.

Overall, our findings move beyond the assumption that "longer is better" and establish a principled foundation for calibrating CoT generation. By adapting to the optimal CoT length, we can develop LLMs that reason more effectively, avoiding both underthinking and counterproductive overthinking.

## 2 UNDERSTANDING CoT LENGTHS IN REAL-WORLD LLMs

To ground our investigation in practical scenarios, we first examine the relationship between CoT length and reasoning performance in publicly available LLMs, and then study how reinforcement learning (RL) influences this relationship. We evaluate the Qwen2.5 series of Instruct models (Qwen et al., 2025) on the MMLU STEM benchmark, which contains challenging competition-level science and engineering problems (Hendrycks et al., 2021a). For each question, we generate 60 solutions spanning a wide range of lengths, where **CoT length is measured by the number of intermediate reasoning steps**. The *optimal CoT length* is defined as the one that yields the highest average accuracy. Additional details on step segmentation and length control are reported in Appendix D. To ensure diversity, we also consider tasks from **mathematics** (MATH), **science** (MMLU STEM), and **commonsense reasoning** (WinoGrande) across four different Qwen2.5-Instruct model sizes, though for clarity we present the MMLU STEM results in the main text and defer the rest to Appendix F.

**Optimal Length Decreases with Stronger Model Capabilities:** As depicted in Figure 2a, there is a clear trend where the optimal CoT length decreases as the model size increases. For instance, the optimal length shifts from 11, 10 steps for the 1.5B and 7B parameter model to 3, 4 steps for the 32B and 72B parameter model. This suggests that more capable models can consolidate reasoning into fewer, more potent steps, aligning with the Simplicity Bias concept where stronger models prefer shorter effective paths.

**Optimal Length Grows with Harder Tasks:** We also investigate how task difficulty influences the optimal CoT length. We use (1 - accuracy) on these questions as a proxy for the difficulty. Figure 2b shows a statistically significant positive correlation (notably $p = 1 \times 10^{-4} \ll 0.05$) between task difficulty and the optimal CoT length of Qwen2.5-7B-Instruct model. This indicates that more challenging problems will significantly benefit from a longer CoT with more extended decomposition steps. Similar trends for other models are provided in Appendix F.1.

**RL does not always yield longer CoTs.** A common belief in the development of advanced reasoning models is that reinforcement learning (RL) naturally produces longer reasoning traces. However, recent studies (Gandhi et al., 2025) suggest that the effect of RL on CoT length is strongly tied to the underlying base model, and that observed increases in length may reflect phenomena such as backtracking rather than genuinely deeper reasoning. To better understand this process, we monitor the evolution of CoT length during GRPO training (Shao et al., 2024) on LeetCode-2K (Xia et al., 2025) with Qwen2.5-7B-Instruct (Qwen et al., 2025). As shown in Figure 2c, optimizing outcome-based rewards can actually reduce the average response length as training converges. Consequently, the RL-trained model produces shorter CoTs than its base counterpart on average, indicating that RL can exert mixed and non-monotonic influences on CoT length.

## 3    A CONTROLLED STUDY OF COT LENGTH ON SYNTHETIC DATASETS

The real-world CoTs usually involve numerous uncontrolled variables (e.g., diverse reasoning strategies, planning, backtracking) and heterogeneous pre-training of base models, which makes precise mechanistic understanding difficult. To overcome these limitations and rigorously examine our hypotheses about optimal CoT length and Simplicity Bias, we design controlled synthetic experiments.

### 3.1    EXPERIMENTAL SETUP

**A Simple Arithmetic Problem.** Our first synthetic dataset consists of arithmetic problems involving sequences of addition operations. The intrinsic difficulty of a problem is quantified by the total number of addition operators, $T$. For each problem with $T$ operators, we construct multiple valid CoT solutions that differ in length and granularity. The CoT length $N$ is defined as the number of intermediate reasoning steps, where each step $i$ processes $t_i$ operators. For simplicity in this controlled study, we enforce $t_i \approx t$ across steps, where $t$ denotes the step size (operators per step) and $N \approx T/t$.

For example, consider the problem `"1+2+3+4+5+6+7"`, which contains $T = 6$ addition operators. We can construct different CoT solutions:

- A *long CoT solution* with $t = 1$ (one operator per step), yielding $N = 6$ steps:

    ```
    Problem: 1+2+3+4+5+6+7
    Step 1: 1+2 = 3. (Remaining: 3+3+4+5+6+7)
    Step 2: 3+3 = 6. (Remaining: 6+4+5+6+7)
    ...
    Step 6: 21+7 = 28. (Final Answer)
    ```

- A *shorter CoT solution* with $t = 3$ (three operators per step), yielding $N = 2$ steps:

    ```
    Problem: 1+2+3+4+5+6+7
    Step 1: 1+2+3+4 = 10. (Remaining: 10+5+6+7)
    Step 2: 10+5+6+7 = 28. (Final Answer)
    ```

This dataset design enables systematic variation of CoT length ($N$) and step size ($t$) for problems with fixed total difficulty ($T$). It allows us to isolate how the structure of the reasoning process itself influences performance. Additional details on problem formulation, data format, and CoT generation are provided in Appendix B.

**A Dynamic Programming Problem.** Beyond arithmetic tasks, we also consider a more general dynamic programming (DP) setting: the *Maximum Path Sum in a Number Triangle*, as studied in prior CoT theory (Feng et al., 2023). The objective is to find a path from the apex to the base that maximizes the sum. The canonical bottom-up DP algorithm solves this by iteratively updating values from the second-to-last row upward. By varying how many rows are merged in each update, we can directly control the effective CoT length while still guaranteeing correctness. This property makes the problem naturally decomposable into solutions of different lengths, closely mirroring the arithmetic case. Results on this DP task are consistent with our observations on arithmetic addition, further

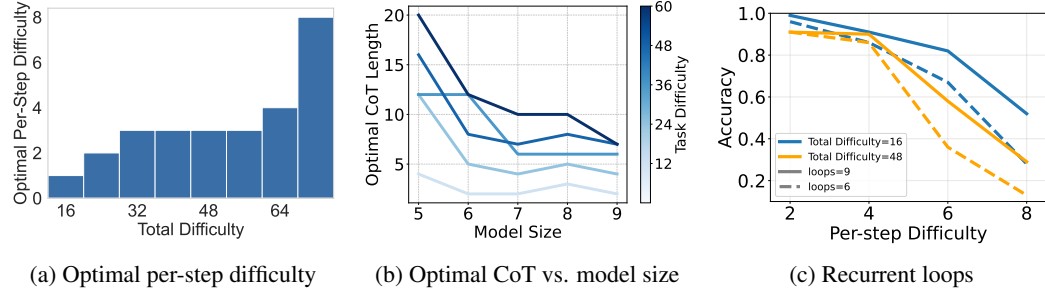

(a) Optimal per-step difficulty       (b) Optimal CoT vs. model size       (c) Recurrent loops

Figure 3: **CoT behaviors in synthetic experiments.** (a) Optimal per-step difficulty ($t$) increases with task difficulty, indicating that harder tasks requires solving more complex sub-steps. (b) Optimal CoT length decreases with larger model size, while harder tasks require longer optimal CoTs at any fixed size. (c) Accuracy comparison between 6-loop and 9-loop models across task difficulties. As per-step difficulty grows, the performance gap widens, indicating that allocating additional loops, i.e., greater per-step reasoning effort, provides clear benefits on harder sub-problems.

reinforcing the generality of the phenomena. For brevity, we focus on the arithmetic results in the main text and defer the DP experiments and details to Appendix C.

**Model and Training:** We train GPT-2 models (Radford et al., 2019) of varying depths (number of layers), keeping other hyperparameters fixed. Model depth is known to be a significant factor representing model capabilities for reasoning tasks (Ye et al., 2024; Chen et al., 2024a). Controlling this hyperparameter alone allows us to study the impact of model capability on optimal CoT length. Models are trained with CoT solutions that can be automatically synthesized for this task, with varying total operators $T$ and CoT lengths $N$ (or equivalently the step sizes $t$). For testing, we can guide the model to produce a CoT of a specific length (e.g., by prompting with a control token indicating the desired number of operators $t$ per step) or allow it to choose its preferred length. Further details are in Appendix H.

## 3.2    Scaling Laws of the Optimal CoT Length and Practical Insights

Our controlled experiments not only corroborate the CoT behaviors observed in real-world scenarios but also allow for a more fine-grained analysis. These findings uncover several key scaling behaviors of the optimal CoT length that shed light into the practical designs of LLM reasoning.

**I. Harder-Tasks' CoTs Peak at Longer Lengths (Adaptive CoT Length Matters):** Our synthetic experiments further confirm the existence of an optimal CoT length, which manifests itself as an inverted U-shaped performance curve when plotting accuracy against the number of reasoning steps, as shown in Figure 1a and 1b. This clearly indicates that both "underthinking" (CoT too short) and "overthinking" (CoT too long) are detrimental, underscoring the critical benefit of generating CoTs with adaptive lengths tailored to the problem's demands. Moreover, we observe that the optimal CoT length shifts right as the task difficulty $T$ gets larger, indicating that solving a harder task optimally requires a longer CoT (also observable numerically from Figure 3b). This suggests that a good reasoning model should be able to vary CoT lengths w.r.t. the overall task complexity.

**II. Harder Tasks Peak at Harder Sub-tasks (Adaptive Per-Step Computation Helps):** Figure 3a illustrates how the number of operators per step ($t$) impacts accuracy across different task difficulties ($T$). The envelope curve, tracing peak performance, shows that as tasks become harder (larger $T$), optimal accuracy is often achieved by CoTs that involve more complex computations *per step* (i.e., a larger optimal $t^*$). This indicates that for difficult problems, simply increasing the number of short, simple steps is insufficient—effective reasoning also requires increasing the complexity of the sub-tasks addressed at each step.

**Implication on Model Choice.** Current LLMs, with fixed Transformer depth, have limited ability to adapt their per-step computation, which constrains their reasoning strategies. In contrast, recent designs such as looped Transformers, which allow adaptive recurrent depth (Geiping et al., 2025; Chen et al., 2025), provide a mechanism to dynamically adjust per-step reasoning effort. This property directly aligns with the observed need for adaptive per-step computation.

To further validate this, we study looped Transformers where the same model can allocate more recurrent loops to increase reasoning effort at each step (Appendix E). Figure 3c plots accuracy against CoT length under fixed task complexity. As per-step difficulty increases, the performance gap between using 6 loops and 9 loops widens, showing that models benefit from allocating more reasoning effort (loops) when sub-tasks are harder. This finding highlights the importance of adaptive reasoning depth: looped Transformers should be trained not only to handle longer CoTs but also to adjust their per-step computation according to task difficulty. To our knowledge, this direction remains underexplored but offers substantial potential for advancing reasoning performance.

**III. Stronger Models Achieve Optimal Performance with Shorter CoTs (Model-Aware CoT Data Matter):** We also examine how model capability (number of layers) influences the optimal CoT length. Figure 3b indicates that, across different task complexities, the optimal number of CoT steps ($N^*$) consistently decreases as the model's capability (number of layers) increases. This is because stronger models can effectively handle more complex sub-tasks in each step, thus requiring fewer overall steps to reach the solution optimally. This finding has significant implications for training data curation. It suggests that to achieve peak performance, models of different sizes or capabilities require CoT data tailored to their respective optimal per-step complexities. Current practices, such as using the same CoT datasets to train LLMs of varying sizes or directly distilling CoTs from large models to small ones without adapting complexity, may be suboptimal. For instance, a small model might struggle to learn effectively from overly complex CoT demonstrations designed for a larger model. Our analysis advocates for training each model with CoT data of adaptive complexity, aligned with its specific capabilities, to help it reach its optimal reasoning performance.

**IV. RL Training Converges to Optimal CoT Length (RL Calibrates Reasoning Behaviors):** As discussed in Section 2, RL training of LLMs tends to shorten CoT length. Our synthetic experiments replicate this effect. Starting from a GPT-2 model pre-trained on CoT solutions of mixed lengths for a task with difficulty $T = 24$, we apply RL with rule-based outcome rewards using PPO on VERL (Schulman et al., 2017; Sheng et al., 2025). In our synthetic setup, each question comes with demonstrations of multiple CoT lengths, so the model naturally develops its own length preference. Figure 4 tracks how this preference shifts under RL, as reflected by the probability (shown on the y-axis) that the model spontaneously selects CoTs of different lengths. The base model spreads probability across lengths around 5, 12, and 24, but as RL training progresses, this distribution shifts to shorter CoTs and finally collapses toward the length 5, which is the accuracy-optimal length under controlled evaluation.

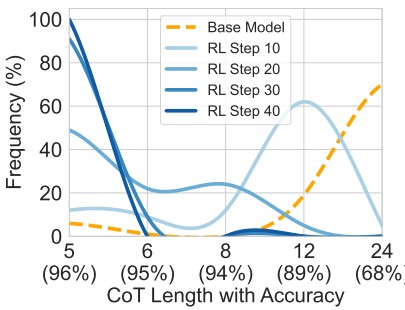

Figure 4: Change of CoT length during RL on the arithmetic task. RL gradually shifts CoTs towards the optimal length ($t^* = 5$), which is shorter than the initial CoTs in average.

This demonstrates that RL, by directly optimizing task success, implicitly steers the model's CoT generation policy toward the optimal length regime, thereby calibrating mismatches between training data and task-model requirements. From this viewpoint, the benefit of RL in LLM training extends beyond reward shaping or exploration: it also serves as an adaptive mechanism for aligning reasoning length. Even when the CoT data used in supervised learning is suboptimal (e.g., misaligned with task complexity or model capability), RL can automatically adjust the model's behavior toward generating more effective, optimally sized CoTs.

**V. Self-Correction Training Shortens Optimal CoTs by Hardening Per-Step Reliability (Error-Tolerant Reasoning Improves Efficiency):** To study how self-correction interacts with optimal CoT length, we modify the training traces so that the model occasionally encounters an intentionally incorrect intermediate result, immediately followed by a local repair. Concretely, instead of exposing the model only to clean chains of the form *question + step$_1$ + **ans**$_1$ + step$_2$ + . . .*, we sometimes replace the first occurrence of a sub-result with a corrupted one and then show the corrected computation: *question + [step$_1$ + **wrong_ans**$_1$] (optionally) + step$_1$ + **correct_ans**$_1$ + step$_2$ + . . . .* We control the fraction of such injected erroneous segments by a parameter $p$ and, through preliminary sweeps, set $p = 0.3$, which strikes a balance between preserving core computational ability and providing sufficient exposure to local error repair. All other training configurations are kept fixed, and we train a 6-layer GPT-2 model under this setting.

Table 1: Optimal CoT length $N^*$ and per-step difficulty $t^*$ with and without self-correction (SC) across task difficulties $T$.

| Task difficulty $T$ | 16 | 24 | 32 | 40 |
|---|---|---|---|---|
| Optimal CoT length $N^*$ w/o SC | 4 | 5 | 8 | 10 |
| Optimal CoT length $N^*$ w/ SC | 2 | 2 | 3 | 5 |
| Optimal subtask difficulty $t^*$ w/o SC | 4 | 5 | 4 | 4 |
| Optimal subtask difficulty $t^*$ w/ SC | 8 | 12 | 11 | 8 |

At test time, we evaluate the model on arithmetic tasks of fixed difficulty and vary the CoT length, selecting the optimal length as the one that maximizes accuracy, as in our previous synthetic analyses. Importantly, when the model executes a self-correction within a single logical step, we count the original erroneous computation and its immediate correction as *one* CoT step, since our notion of CoT length reflects how finely the problem is decomposed, not how many times a local computation is revised. We then extract both the optimal number of steps $N^*$ and the corresponding optimal per-step difficulty $t^*$ (operators per step) across task difficulties $T \in \{16, 24, 32, 40\}$.

As shown in Table 1, self-correction training substantially reduces the optimal CoT length across all task difficulties (first two rows), while simultaneously shifting the optimal per-step difficulty to significantly larger values (last two rows). Although the shorter $N^*$ might appear counter-intuitive given the extra "thinking" introduced during training, it is fully consistent with our broader picture: by learning to reliably repair local mistakes, the model becomes more robust to error accumulation within each step, which in turn allows it to tackle harder sub-tasks per step (larger $t^*$) without sacrificing accuracy. From a data-design perspective, these results suggest that injecting structured self-correction signals into CoT training can be an effective way to teach models to use fewer, but more powerful, steps.

## 4    AN ERROR ACCUMULATION ANALYSIS ON CHAIN-OF-THOUGHT

Empirical studies on both real-world and synthetic datasets consistently suggest the existence of an optimal Chain-of-Thought (CoT) length. To explain this, we develop a theoretical model based on an intuitive analysis of accumulated errors and extend it to more general settings. Remarkably, the predictions of this simple model align closely with the empirically observed scaling behaviors of CoT length in our toy model and large language models. While not exhaustive, these insights provide a useful lens for understanding and anticipating how CoT length influences reasoning performance. All proofs are deferred to Appendix J.

**Setup.** Consider the arithmetic task with $T$ operators and an $N$-step CoT as in Section 3.1. At step $i$, the model produces a sub-question $q_i$ and a sub-answer $a_i$, with history $H_{i-1} = [q_1, a_1, \ldots, q_{i-1}, a_{i-1}]$. We use the likelihood factorization

$$P(a_{\text{final}}|q, \theta, N) = \prod_{i=1}^{N} \underbrace{P(q_i|H_{i-1}, q, \theta, N)}_{\text{sub-question}} \underbrace{P(a_i|q_i, H_{i-1}, q, \theta, N)}_{\text{sub-answer}},$$

We abstract diverse "reasoning behaviors" (reflection, verification, backtracking) as particular choices of task decomposition and focus on two error sources: (i) **sub-question error** $\sigma(T) \in [0, 1)$, increasing with difficulty $T$; (ii) **sub-answer error** $E(N, M, T) \in [0, 1]$, depending on model capability $M$ and effective per-step difficulty $T/N$. For each model with parameters $\theta$, we define its capability $M(\theta)$ using the *reasoning boundary* (Chen et al., 2024b):

$$M = M(\theta) = \max_t \{ \Pr(a_i = a_i^* \mid t_i, \theta) > \varepsilon, \ |t_i| = t \},$$

where $|t_i|$ is the number of operators in the subtask $t_i$. Intuitively, $M(\theta)$ represents the largest sub-problem size the model can reliably solve in a single reasoning step.

**Proposition 4.1.** *Assuming stepwise stationarity and independence conditioned on history, the final accuracy takes the form*

$$A(N) = P(a_{\text{final}} = a_{\text{final}}^* \mid q, \theta, N) = \alpha\big((1 - \sigma(T))(1 - E(N, M, T))\big)^N, \qquad (1)$$

*where $\alpha$ denotes a constant independent of $N$.*

**A solvable special case.**  For intuition, consider a linear sub-question error rate $\sigma(T) = \frac{T}{C}$, where C denotes the maximum task difficulty the model family is trained to handle, which is the largest operator count present in the training distribution (with $T/C \leq 0.9$ within the training regime). Similarly, assume a linear sub-answer error $E(N, M, T) = T/(NM)$, which captures the number of operators processed per step relative to the model's capability $M$. Then

$$A(N) = \alpha\big(1 - \tfrac{T}{C}\big)^N \big(1 - \tfrac{T}{NM}\big)^N, \tag{2}$$

which increases for small $N$ (decomposition helps) and decreases for large $N$ (errors accumulate).

**Theorem 4.2** (Optimal CoT length)**.** *There exists an optimal $N^*(M, T)$ maximizing $A(N)$:*

$$N^*(M, T) = \frac{T\,Z}{M(Z + 1)}, \quad Z = W_{-1}\Big(-\Big(1 - \tfrac{T}{Ce}\Big)\Big),$$

*where $W_{-1}$ is the negative branch of the Lambert W function ($we^w = x$).*

This theorem establishes the inverted U-shaped relationship between CoT length and accuracy, and provides an explicit formula for the optimal length $N^*$. From this expression, we can formally derive the first three scaling behaviors characterized in Section 3.2.

**Corollary 4.3** (Scaling laws)**.** *From Theorem 4.2:*

- *$N^*(M, T)$ increases with $T$ (harder tasks warrant longer CoT).*

- *The optimal operators per step $t^* = T/N^*(M, T) = M(1 + 1/Z)$ increases with $T$ (envelope behavior).*

- *$N^*(M, T)$ decreases with $M$ (stronger models need fewer steps).*

**How RL Calibrates CoT.**  The same analysis also sheds light on why reinforcement learning (RL) with outcome supervision help calibrates CoT length (Section 3.2). During RL, the choice of CoT length can be viewed as selecting an action $N_i$ from a discrete set $\mathcal{A} = \{N_1, \ldots, N_k\}$. Each $N_i$ produces a binary reward $r \in \{0, 1\}$ with success probability $A(N_i)$ from Proposition 4.1, reducing the setting to a stateless bandit. With a softmax policy $\pi_\theta(N_i) = \frac{e^{\theta_i}}{\sum_j e^{\theta_j}}$, the RL objective is

$$J(\theta) = \sum_{i=1}^{k} \pi_\theta(N_i)\, A(N_i), \qquad \nabla_{\theta_i} J = \sum_{j=1}^{k} A(N_j)\pi_\theta(N_j)(\delta_{ij} - \pi_\theta(N_i)).$$

**Corollary 4.4** (RL Converges to Optimal CoT Length)**.** *For gradient ascent on $J(\theta)$ with sufficiently small step size, the policy converges to a deterministic solution $\pi_\theta(N_i) = 1$ iff $i = \arg\max_j A(N_j)$. Thus, RL training converges to the optimal CoT length $N^* = \arg\max_{N \in \mathcal{A}} A(N)$.*

This result shows that RL will automatically prefer the optimal length and hence calibrates the CoT length. In this way, our framework unifies the explanation of optimal CoT length, its scaling laws, and RL's calibration effects of reasoning lengths.

**Extension to Nonlinear and Stochastic Error Functions.**  In the analysis above, we adopted a simple linear model with a closed-form solution for the optimal length to provide intuitive understanding. This framework can be extended to more general settings, including **nonlinear error functions** that are monotone and convex, as well as **stochastic error models** where each subtask may exhibit a different error rate. These extensions introduce additional technical subtleties but follow the same underlying principles. Overall, it shows that the accumulative error analysis can explain a broad class of reasoning process, including the arithmetic and dynamic programming problems we covered in Section 3. Due to space limitations, we defer the formal treatment to Appendix I.

## 5 PRACTICAL APPLICATIONS OF OPTIMAL COT LENGTH

Guided by the understanding above, in this section, we illustrate via some proof-of-concept experiments that adapting LLM training and inference configurations to the optimal CoT length can improve the model's reasoning performance.

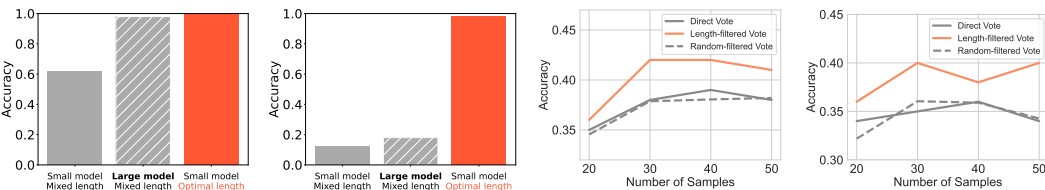

(a) Influence of CoT Data ($T = 32$)

(b) Influence of CoT Data ($T = 64$)

(c) Length-Filtered Vote (Llama3-8B-Instruct)

(d) Length-Filtered Vote (Qwen2.5-7B-Instruct)

Figure 5: (a) and (b) compare model performance under different pretraining data distributions: Mixed Length (uniform over all lengths) vs. Optimal Length (only optimal-length solutions). Despite its smaller size, the small (6 layer) model trained on optimal-length data outperforms the large (9 layer) model trained on mixed-length data, with the performance gap widening as task difficulty increases. (c) and (d) validate our Length-Filtered Vote method on different models, which consistently outperforms vanilla majority vote and random-filtered vote on the GPQA dataset, maintaining strong performance even as the number of samples increases.

## 5.1 TRAINING WITH DATA OF OPTIMAL CoT LENGTH

**Training with Optimal-Length CoT Data:** The existence of an adaptive, optimal CoT length suggests that one should design the CoT training data adaptively to fully optimize the model's reasoning performance. To examine the influence of the CoT length of the training data, we train a model on a specialized dataset that contains CoT solutions with lengths known to be optimal for the given model size and task difficulty ($T$). We compare this model against a baseline model trained on a dataset of CoT solutions with uniformly distributed step lengths $t$. During testing, models were allowed to freely choose their CoT strategy.

**Results.** As shown in Figures 5a and 5b, the model trained on optimal-length CoTs significantly outperforms the models trained on mixed-length solutions. Remarkably, a smaller model (e.g., 6 layers) trained on optimal-length data can even outperform a larger model (e.g., 9 layers) trained on randomly chosen CoT lengths. This proof-of-concept experiment underscores the critical influence of the suitability of the CoT length in training data for the model. While it is generally hard to exactly estimate optimal CoT lengths in real-world problems, our theoretical and empirical studies provide valuable guidelines for a coarse estimate. We leave more in-depth studies to future work.

## 5.2 ADAPTIVE LENGTH-FILTERED VOTE AT INFERENCE TIME

The observation that CoTs of optimal length yield higher accuracy suggests that inference-time strategies could benefit from this insight. Standard approaches like majority voting over multiple sampled CoTs, such as self-consistency (Wang et al., 2023), treat all valid reasoning paths equally, regardless of their length. However, paths that are too short (underthinking) or too long (overthinking and error-prone) may contribute noisy or incorrect answers to the voting pool.

While Fu et al. (2023) previously proposed filtering out short CoTs, it worked for smaller 2023-era models where "longer is better." Inspired by our findings, we propose **Length-Filtered Vote**, an adaptive method that enhances standard majority voting by preferentially weighting or exclusively considering answers derived from CoTs whose lengths fall within a proper range. Specifically, in majority vote, given a model $f_\theta$, a question $q$, a ground truth answer $a^*$, we first sample a set of answer candidates $c_1, \ldots, c_n \overset{i.i.d.}{\sim} f_\theta(q)$ independently. After that, instead of a direct vote, we group the answers by their corresponding CoT length $\ell(c_i)$ (discussed in Appendix D) into groups with equal bin size $D$ (by default, we set $D = 2$), denoted as $\{L_j\}_{j=1}^m$. As our theory suggests that the prediction accuracy is peaked around a certain range of CoT length, we identify such groups through the prediction uncertainty of the answers within each group, based on the intuition that lower uncertainty implies better predictions. Specifically, we calculate the Shannon entropy $H(L_i)$ of the final answers given by the CoT chains in each group $L_i$. We use a majority vote only for the $K$ (by default, we set $K = 3$) groups with the smallest entropy. For **Random-filtered Vote**, we do a random grouping of samples (i.e., not based on length), and repeat the same entropy-based filtering process,

averaging the final results over 100 trials to account for the randomness of the grouping. A detailed description of the algorithm is in Appendix K.

**Results.** We evaluate the proposed method against vanilla majority vote (i.e., self-consistency (Wang et al., 2023)) on a randomly chosen subset from the GPQA dataset (Rein et al., 2023). The results in Figure 5c and 5d show that our filtered vote consistently outperforms vanilla majority vote and random-filtered vote and shows little performance degradation as the sample number increases. In summary, our findings show that **CoT length, as one of the most easily computable feature in scenarios where token-level probabilities are unavailable, is correlated with the final accuracy**.

## 6 RELATED WORK

**Chain-of-Thought for LLM Reasoning.** CoT has become a core technique for LLMs to solve complex reasoning tasks by generating intermediate steps (Wei et al., 2022). Numerous variants arise to enhance CoT reasoning with more structural substeps, such as least-to-most prompting (Zhou et al., 2023), tree of thoughts (Yao et al., 2023), and divide-and-conquer methods (Zhang et al., 2024; Meng et al., 2024). These methods fundamentally treat CoT as a framework for task decomposition and subtask solving that falls in our analysis in Section 4.

**Overthinking in CoT Reasoning.** With the rise of powerful reasoning models like OpenAI o1, scaling test-time compute with long CoT has gained prominence (Snell et al., 2024; Chen et al., 2024d; Wu et al., 2024; Brown et al., 2024). These studies often suggest that more computation like longer CoT can lead to better results. However, this is not always true. With similar interests as ours, a few concurrent works also investigated the "overthinking" phenomenon (Chen et al., 2024c) where reasoning models generate excessively long CoTs for simple problems and proposed some mitigation strategies Han et al. (2024); Luo et al. (2025); Ma et al. (2025); Sui et al. (2025). Our analysis goes beyond these observations by formally establishing the existence of an optimal CoT length and its scaling behaviors. Supported by both controlled experiments and theoretical analysis, it offers principled guidelines for designing more effective CoT strategies.

**Theoretical Understanding of CoT.** Numerous studies aim to theoretically formalize the Chain-of-Thought (CoT) process and understand its effectiveness. They include analyzing CoT's computational advantages via circuit complexity (Feng et al., 2023; Li et al., 2024), and quantifying step-wise information gain from an information-theoretic standpoint (Ton et al., 2024). While Schaeffer et al. (2023) uses error accumulation to explain emergent abilities via a monotonic $p^L$ formulation with fixed reasoning length, our work is the first to leverage error accumulation to analyze the influence of CoT length. In addition, Bao et al. (2024) and FU et al. (2025) identify and characterize the latent causal structures and robustness of model reasoning. Ye et al. (2024) conducted controlled synthetic experiments to help uncover underlying problem-solving mechanisms in LLMs. While Jiang et al. (2025) presents an automated framework that converts sequential Long CoTs into hierarchical tree structures. Distinct from these varied theoretical explorations, our findings on CoT scaling behaviors and the consequent need for model-specific CoT structures (as discussed in Section 3.2) resonate with the concept of algorithmic alignment (Xu et al., 2019), which suggests that models perform best when the problem structure aligns with their computational structure.

## 7 CONCLUSION

This work revisits a prevailing assumption in reasoning with large language models: that longer Chain-of-Thoughts (CoTs) are always better. Through controlled experiments and theoretical analysis, we showed that accuracy instead follows an inverted U-shaped curve with respect to CoT length, revealing the existence of an optimal length that balances finer task decomposition against compounding errors. Our systematic study further uncovered scaling behaviors of this optimal length across task difficulty, model size, per-step computation, and RL training.

Building on these insights, we demonstrated that training with optimally sized CoTs improves performance, and introduced *Length-Filtered Vote* as an effective inference strategy. Together, these findings highlight the importance of calibrating reasoning length rather than adopting a one-size-fits-all approach. We advocate for a principled framework in which LLMs adaptively allocate the right amount of reasoning effort, ultimately leading to more reliable and efficient problem solving.

ACKNOWLEDGMENTS

Yisen Wang was supported by National Natural Science Foundation of China (62376010, 92370129), Beijing Major Science and Technology Project under Contract no. Z251100008425006, Beijing Nova Program (20230484344, 20240484642), and State Key Laboratory of General Artificial Intelligence. Yifei Wang and Stefanie Jegelka were supported in part by the NSF AI Institute TILOS (NSF CCF-2112665), and an Alexander von Humboldt Professorship.

ETHICS STATEMENT

This work complies with the ICLR Code of Ethics. While our methods are general, they may be applied in contexts with societal implications, including risks related to bias, fairness, and privacy. We encourage responsible use and declare no conflicts of interest.

REPRODUCIBILITY

We provide detailed descriptions of our methodology, datasets, model configurations, and evaluation metrics in both the main text and the Appendix. In addition, the complete source code is included in the supplementary materials to facilitate reproducibility.

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

# Appendix

## A   THE USE OF LARGE LANGUAGE MODELS (LLMS)

In this work, LLMs are primarily employed for two purposes: (1) polishing the language of the manuscript to ensure grammatical correctness and coherence, and (2) assisting in the standardized organization and documentation of the released codebase. Importantly, all conceptual development, theoretical analysis, experimental design, and result interpretation are conducted independently by the authors. The use of LLMs is strictly limited to auxiliary tasks, ensuring that the scientific contributions of this paper remain entirely unaffected by such tools.

## B   FORMAL DEFINITIONS OF SIMPLIFIED ARITHMETIC PROBLEM

To begin, we aim to empirically investigate the relationship between reasoning performance and CoT length. Therefore, we need to control a given model to generate reasoning chains of varying lengths for a specific task. Unfortunately, no existing real-world dataset or model fully meets these strict requirements. Real-world reasoning tasks, such as GSM8K or MATH (Cobbe et al., 2021; Hendrycks et al., 2021b), do not provide multiple solution paths of different lengths, and manually constructing such variations is challenging. Moreover, it is difficult to enforce a real-world model to generate a diverse range of reasoning paths for a given question. Given these limitations, we begin our study with experiments on synthetic datasets.

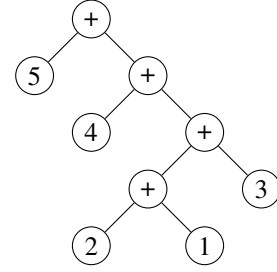

Figure 6: Computation tree of arithmetic expression $5 + (4 + ((2 + 1) + 3))$.

### B.1   PROBLEM FORMULATION

To investigate the effect of CoT length in a controlled manner, we design a synthetic dataset of simplified arithmetic tasks with varying numbers of reasoning steps in the CoT solutions.

**Definition B.1** (Problem). In a simplified setting, an arithmetic task $q$ is defined as a binary tree of depth $T$. The root and all non-leaf nodes are labeled with the $+$ operator, while each leaf node contains a numerical value (mod 10). In addition, we impose a constraint that every non-leaf node must have at least one numerical leaf as a child.

The bidirectional conversion method between arithmetic expressions and computation trees is as follows: *keeping the left-to-right order of numbers unchanged, the computation order of each "+" or tree node is represented by tree structure or bracket structures.* For example, consider the task $5 + (4 + ((2 + 1) + 3))$ with $T = 4$. The corresponding computation tree is defined as Figure 6.

To ensure that CoT solutions of the same length have equal difficulty for a specific problem, we assume that each reasoning step performs the same operations within a single CoT process.

**Definition B.2** (Solution). We define a $t$-hop CoT with a fixed each step length of $t$ as a process that executes $t$ operations starting from the deepest level and moving upward recursively.

According to this definition, the execution sequence is uniquely determined. For example, one way to solve expression in Figure 6 is by performing one addition at a time:

$$5 + (4 + ((2 + 1) + 3)) = \texttt{<1>} \tag{3}$$
$$2 + 1 = 3 \tag{4}$$
$$3 + 3 = 6$$
$$4 + 6 = 0$$
$$5 + 0 = 5\texttt{<END>}.$$

Another approach is to perform two additions at a time:

$$5 + (4 + ((2 + 1) + 3)) = \texttt{<2>} \tag{5}$$
$$(2 + 1) + 3 = 6$$
$$5 + (4 + 6) = 5\texttt{<END>}.$$

The latter approach is half as long as the former, but each reasoning step is more complex[1]. This illustrates a clear trade-off between the difficulty of each subtask and the total number of reasoning steps.

In practice, when $t$ does not evenly divide $T$, the final step performs $T \mod t$ operations. To guide the model in generating the desired CoT length, we insert the control token $\texttt{<t>}$ after the question and before the beginning of the solution. To preserve the parentheses that indicate the order of operations, we construct expressions in Polish notation. However, for readability, we present each problem in its conventional form throughout the article.

### B.2 Contrast to vanilla arithmetic problem

**Why pruning?** Initially, we intended to create a synthetic dataset for regular arithmetic tasks, but we quickly realized that the computation tree for such tasks is uncontrollable. For example, consider the task $1 * 2 + 3 * 4$. We hoped to compute 2 operators in one step, but found it impossible because the addition needs to be computed after the two multiplications, and we cannot aggregate two multiplications in one subtask. Therefore, pruning the computation tree becomes essential.

**Why only focusing on addition?** There are two reasons why we focus on arithmetic tasks involving only addition: first, it simplifies pruning, as the order of operations can be controlled solely by parentheses; second, it facilitates the computation of sub-tasks, since parentheses do not affect the final result, and the model only needs to compute the sum of all the numbers when solving a sub-task. We aim for the model to handle longer sub-tasks, thereby allowing a broader study of the impact of CoT length.

**Will the simplified synthetic dataset impact the diversity of the data?** We need to clarify that even with pruning, the structure of the expressions will still vary because swapping the left and right child nodes of each non-leaf node in the computation tree results in different expressions. When $T > 30$, the number of possible variations exceeds $1 \times 10^9$.

## C Dynamic Programming (DP) Problems

### C.1 Experimental Setup

To complement the arithmetic dataset, we design a classical dynamic programming (DP) problem — the **Maximum Path Sum in a Number Triangle**. This task shares the same desirable decomposability property as the arithmetic problems: it naturally admits multiple solutions of varying CoT lengths, making it suitable for analyzing the scaling behavior of reasoning length.

We construct a dataset of number triangles with varying heights $H$. Each triangle consists of $H$ rows, where the $i$-th row contains $i$ integers sampled uniformly from a predefined range (e.g., $[1, 99]$). The

---

[1]This is because performing two operations at once requires the model to either memorize all combinations of numbers in a two-operator equation and their answers, apply techniques like commutativity to reduce memory requirements, or use its mental reasoning abilities to perform the two operations without relying on CoT.

total task difficulty is quantified by the number of rows $H$, since longer triangles require deeper reasoning chains to propagate information from the base to the top.

For example, consider the following triangle of height $H = 4$:

$$
\begin{array}{ccccccc}
& & & 7 & & & \\
& & 3 & & 8 & & \\
& 8 & & 1 & & 0 & \\
2 & & 7 & & 4 & & 4
\end{array}
$$

The goal is to find a path from the apex (top) to the base that maximizes the sum of visited numbers. The canonical solution employs a bottom-up dynamic programming algorithm: starting from the second-to-last row, we update each entry as the sum of the current value and the maximum of its two children in the row below. Repeating this process row by row eventually yields the maximum path sum at the apex.

A long CoT solution might be designed to process $t = 1$ layer per step.

$$
\begin{aligned}
\text{Row 3 update: } & [\, 8 + \max(2, 7), \; 1 + \max(7, 4), \; 0 + \max(4, 4) \,] \\
& = [\, 15, \; 8, \; 4 \,] \\
\text{Row 2 update: } & [\, 3 + \max(15, 8), \; 8 + \max(8, 4) \,] \\
& = [\, 18, \; 16 \,] \\
\text{Row 1 update: } & [\, 7 + \max(18, 16) \,] \\
& = [\, 25 \,]
\end{aligned}
$$

A shorter CoT solution for the same problem might process $t = 2$ layers per step.

$$
\text{Step 1 (Row 4} \to \text{Row 2): }
\begin{cases}
\text{For Row 2, Col 1: } 3 + \max(2 + 8, \; 7 + 8) = 3 + \max(10, \; 15) = 18 \\
\text{For Row 2, Col 2: } 8 + \max(7 + 1, \; 4 + 1) = 8 + \max(8, \; 5) = 16
\end{cases}
$$

$$
\Rightarrow \text{Row 2 becomes } [\, 18, \; 16 \,]
$$

Step 2 (Row 2 $\to$ Row 1): $7 + \max(18, \; 16) = 7 + 18 = 25$

Thus, the maximum path sum is 25.

## C.2   EVALUATION OF DYNAMIC-PROGRAMMING TASKS ON LARGER QWEN2.5 MODELS

To further validate the non-triviality of our dynamic-programming (DP) benchmark and examine its behavior on stronger models, we evaluate Qwen2.5 instruct models of varying sizes on DP tasks of depths 6, 8, and 10. For each difficulty level, we generate 100 problem instances and sample 10 CoT responses per instance. Following our real-world evaluation protocol, we determine the *optimal CoT length* for each model–task pair by selecting the chain length that achieves highest accuracy.

We omit the 1.5B model from analysis due to its low performance (<10% accuracy). Importantly, even the 72B model does not achieve perfect accuracy on depth-6 tasks (81.8%), indicating that our DP benchmark remains *non-trivial* and effectively probes structured algorithmic reasoning.

Table 2 summarizes optimal CoT lengths and corresponding accuracies. Two clear trends emerge: 1) larger models consistently require *shorter* optimal CoT lengths, and 2) deeper DP tasks require *longer* optimal CoT lengths. Both findings are fully aligned with our theoretical predictions regarding adaptive CoT behavior with respect to model capability and task difficulty.

## D   SUPPLEMENTARY DETAILS ON REAL WORLD EXPERIMENT FOR OPTIMAL COT LENGTH

### D.1   SOLUTION LENGTH CONTROL

To study the impact of CoT length on performance under a given problem difficulty, we need to induce the model to naturally generate solutions of varying lengths. Simply adding prompts like

Table 2: Optimal CoT length (accuracy %) across Qwen2.5 model sizes and DP task depths.

| DP Depth | 6 | 8 | 10 |
|---|---|---|---|
| 7B | 5 (35.0%) | 7 (30.3%) | 8 (22.6%) |
| 32B | 4 (70.6%) | 5 (45.8%) | 8 (42.9%) |
| 72B | 3 (81.8%) | 5 (66.7%) | 6 (46.2%) |

"*please use 100 tokens to solve this problem*" or "*please use 10 steps to solve this problem*" is not ideal because the model's ability to follow instructions regarding output length is limited, and such fixed-length prompts may not ensure fairness across problems of different difficulties. Moreover, prompting for a specific length might lead the model to generate irrelevant tokens or steps just to "pad the length," without actually changing the number of steps or the complexity of the reasoning. Additionally, controlling max_length is also problematic, as overly long responses might get truncated, which would directly lead to lower accuracy for longer outputs. What we really want is for the model to generate a complete and coherent long response on its own, so we can observe the corresponding accuracy.

To create solutions with varying step lengths with different complexity, we follow (Fu et al., 2023) by using in-context examples (8-shots) with three different levels of complexity to guide the model in generating solutions with different step counts. For each set of in-context examples, we sample 20 times, resulting in a total of 60 samples per question.

## D.2 STEP SEGMENTATION

Simply measuring CoT length by counting tokens is neither rigorous nor meaningful. Since our focus is on final performance rather than efficiency, we care more about using CoT length to reflect the complexity of the reasoning pattern. In this sense, the number of reasoning steps can serve as a more appropriate indicator of CoT length. As we discussed earlier, the step number captures how the model decomposes the problem, which directly reflects the complexity of its reasoning. In contrast, token length fails to capture this because, as the model thinks more deeply and the number of steps increases, the number of tokens per step may decrease—making the total token count unpredictable and unreliable as a proxy for reasoning complexity.

When calculating the number of steps, we separate the full reasoning chain using "\n"(Fu et al., 2023) and remove empty lines caused by "\n\n". Then we consider the total number of lines as the CoT length. Since questions in the MATH dataset are challenging and lead to high variability in final CoT lengths, we normalize the lengths by applying length = length // bin_width. For experiments comparing different models (e.g., optimal CoT length per model or optimal vs. longest CoT), the questions within each length bin differ, which introduces variability. To reduce this variance and ensure each bin has enough samples, we use a relatively large bin width of 5. In contrast, for analyzing the influence of task difficulty, where each calculation on optimal CoT length only contains one question, we adopt a finer bin width of 2 for better resolution (we also verified that using width 1 yields almost identical results).

## D.3 MORE DETAILS OF FIGURE 2B.

When evaluating the results, questions with accuracy $< 0.01$ or $> 0.99$ (indicating all incorrect or all correct responses) are excluded, as their accuracy does not vary with step length changes.

To better understand the reliability of the observed trend between task difficulty and optimal Chain-of-Thought (CoT) length, we compute a 95% confidence interval around the linear regression line. Specifically, we use standard methods based on the Student's t-distribution to estimate uncertainty in the predicted values. The confidence band reflects how much the estimated mean CoT length is expected to vary given the finite sample size and the distribution of data points.

### D.4 On the Definition of Task Difficulty and Mitigating Accuracy Bias

Defining task difficulty requires particular care because raw accuracy can be biased by a model's inherent preference for certain CoT lengths. We address this issue using two complementary strategies.

**Controlling for CoT-Length Bias.** Different questions may elicit different preferred CoT lengths from the model, which can artificially inflate or deflate their measured difficulty. To reduce this confounding factor, for each question we explicitly prompt the model to generate *multiple* CoT lengths when producing the 60 responses used in our evaluation. This procedure minimizes the effect of internal length biases on accuracy and yields a more reliable task-difficulty signal.

**Model-Aware Difficulty Definition.** Task difficulty is inherently model-dependent: a problem that is easy for one model may be challenging for another due to differences in scale, data coverage, or training. For this reason, we define difficulty in a model-aware manner by using the *average accuracy of the evaluated model* as the difficulty indicator. This ensures that the difficulty metric faithfully reflects the model's actual competence rather than relying on externally imposed or model-agnostic notions of hardness.

## E Looped Transformer

Following Bae et al. (2025), we implement the looped Transformer architecture by iteratively applying a single Transformer layer multiple times. Specifically, we train two variants with `loops` = 6 and `loops` = 9, both configured with an embedding dimension of $64 \times 9$ and 9 attention heads (only 4 MB parameters in total). Training is conducted on a mixed dataset with maximum task difficulty = 64 and maximum CoT hop size = 8. Detailed training procedures are provided in the released code.

## F Extended Experiments on Broader Domains

To strengthen the generality of our experimental conclusions, we expanded both the scale of the data and the diversity of reasoning domains considered. Specifically, we conducted experiments in the following three areas:

- **Mathematical reasoning:** We employed the full MATH500 dataset (Lightman et al., 2023). This dataset contains a curated subset of 500 problems from the original MATH benchmark.
- **Scientific reasoning:** We adopted the MMLU STEM (Hendrycks et al., 2021a) dataset, which is a subset of STEM subjects defined in the original MMLU benchmark, which covers a wide range of scientific and engineering domains.
- **Commonsense reasoning:** We used the full Winogrande (Sakaguchi et al., 2019) xs training split. This dataset formulates a fill-in-the-blank task with binary options, designed to require non-trivial commonsense reasoning.

### F.1 Additional Results for Figure 2b.

Before presenting the results on additional datasets, we first further investigate the relationship between task difficulty and optimal CoT length on real-world benchmarks using different models. As shown in Figure 7, the findings are consistent and compelling: across all evaluated models, we observe a clear and statistically significant correlation between task difficulty and the corresponding optimal CoT length. These analyses are also validated on broader datasets such as MATH500, MMLU STEM, and Winogrande.

### F.2 Experiments on the Full MATH500 Dataset

We acknowledge the need for broader validation beyond a single subset of data. Therefore, we further conducted experiments on the complete MATH500 dataset. Specifically, we evaluated the Qwen2.5-Instruct models (1.5B, 7B, 32B, and 72B) with 30 sampled solutions per question.

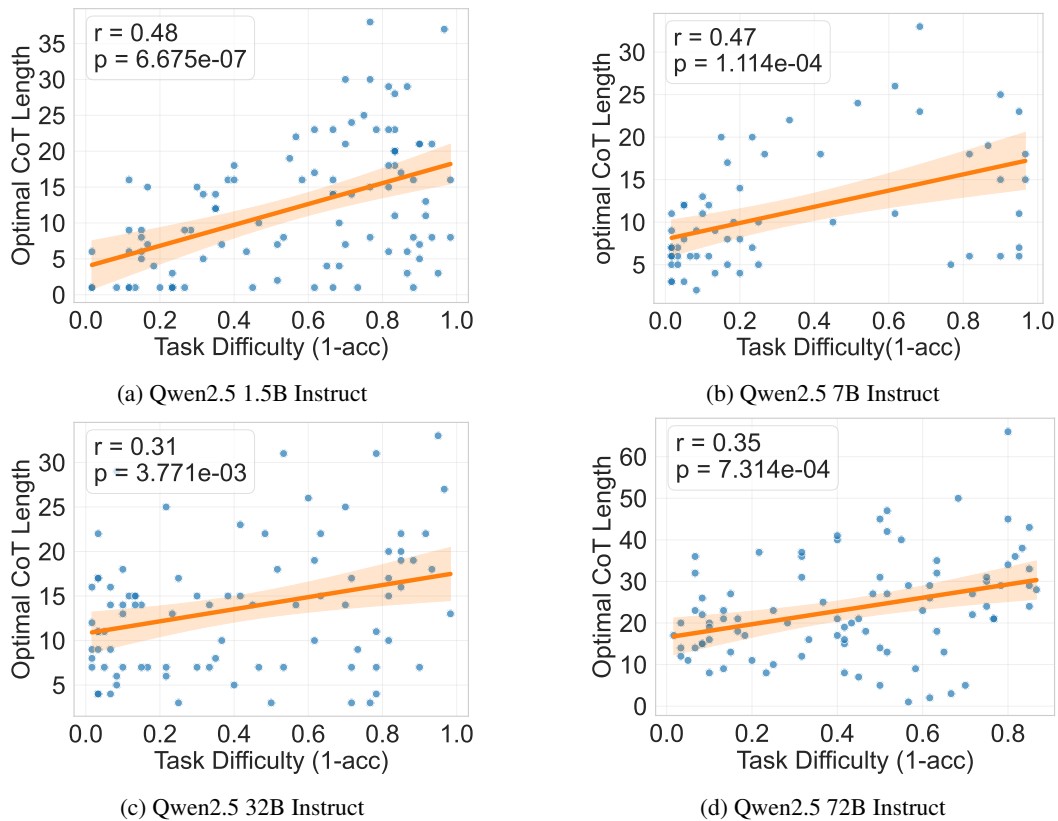

Figure 7: Evaluation between task difficulties and optimal CoT lengths on MMLU STEM datasets.

Table 3 reports the comparison of accuracy achieved with the longest chain-of-thought (CoT) versus the optimal CoT length, across different model sizes. Table 4 further examines the relationship between optimal CoT length and problem difficulty.

Table 3: Optimal CoT length vs. model size on the full MATH500 dataset.

|                            | 1.5B         | 7B           | 32B          | 72B          |
|----------------------------|--------------|--------------|--------------|--------------|
| Accuracy (longest length)  | 0.18         | 0.27         | 0.40         | 0.08         |
| Accuracy (optimal length)  | 0.38 (**+0.20**) | 0.82 (**+0.55**) | 0.81 (**+0.41**) | 0.81 (**+0.73**) |
| Optimal length             | **5**        | 2            | 1            | 2            |

Table 4: Correlation between optimal CoT length and task difficulty.

|       | 1.5B   | 7B     | 32B    | 72B    |
|-------|--------|--------|--------|--------|
| $r$   | 0.2092 | 0.2378 | 0.2266 | 0.1986 |
| $p$   | 0.0068 | 0.0034 | 0.0029 | 0.0297 |

In this experiment, where problem difficulty was not explicitly controlled, we observed that the 7B, 32B, and 72B models achieved peak performance at shorter CoT lengths. This trend is likely explained by the dataset's high concentration of easier problems (levels 1–4), which, as shown in our previous results, generally require shorter reasoning chains. By contrast, the much weaker 1.5B model still benefited from longer reasoning (optimal length = 5).

Importantly, despite the skewed distribution of problem difficulty, the key relationship we aimed to study remains intact: we consistently found a statistically significant correlation ($p < 0.05$)

between task difficulty and optimal CoT length, which aligns with our earlier findings across all other experiments.

### F.3 EXPERIMENTS ON THE WINOGRANDE DATASET

We have also conducted new experiments on the WINOGRANDE (Sakaguchi et al., 2019) dataset to assess commonsense reasoning. In particular, we evaluated the Qwen2.5-Instruct models (1.5B, 7B, 32B, and 72B) on the full WINOGRANDE-XS training split, replicating the experimental setup from Figure 2 of our main paper.

Table 5 reports the comparison between the longest and optimal CoT lengths across different model sizes, while Table 6 shows the correlation between optimal CoT length and task difficulty.

Table 5: Optimal CoT length vs. model size on the WINOGRANDE-XS dataset.

|                           | 1.5B        | 7B          | 32B         | 72B         |
| ------------------------- | ----------- | ----------- | ----------- | ----------- |
| Accuracy (longest length) | 0.56        | 0.69        | 0.72        | 0.85        |
| Accuracy (optimal length) | 0.63 (+0.07)| 0.74 (+0.05)| 0.80 (+0.08)| 0.93 (+0.08)|
| Optimal length            | 15          | 15          | 10          | 9           |

Table 6: Correlation between optimal CoT length and task difficulty on WINOGRANDE-XS.

|     | 1.5B   | 7B       | 32B    | 72B    |
| --- | ------ | -------- | ------ | ------ |
| $r$ | 0.2201 | 0.4256   | 0.3886 | 0.2098 |
| $p$ | 0.0052 | < 1e-4   | 0.0120 | 0.0077 |

These experiments significantly broaden the generalizability of our conclusions. The results corroborate our earlier findings from the mathematical domain, demonstrating that the optimal CoT length decreases as the model size increases and that it remains significantly correlated with task difficulty ($p < 0.05$).

## G SUPPLEMENTARY DETAILS ON REAL WORLD EXPERIMENT FOR RL SIMPLICITY BIAS

For Figure 2c, we use Qwen2.5-7B-Instruct (Qwen et al., 2025) as the base model, Group Relative Policy Optimization with R1-like prompting (Shao et al., 2024; Guo et al., 2025) for the reinforcement learning process, and LeetCode-2K (Xia et al., 2025) as the training and evaluation dataset. We take the following training configuration by default:

Table 7: Hyperparameter settings for real-world RL experiments with Qwen2.5-instruct models.

| Learning Rate | Max Epochs | Rollout Samples | Reverse KL Coefficient | Entropy Loss Coefficient | Effective Batch Size |
| ------------- | ---------- | --------------- | ---------------------- | ------------------------ | -------------------- |
| 5e-7          | 10         | 16              | 1e-3                   | 5e-3                     | 256                  |

## H ADDITIONAL SYNTHETIC EXPERIMENT DETAILS

### H.1 TRAINING DETAILS

In default, we train different models (layers ranging from 5 to 9) on the same dataset, which included mixed questions with total operators $T \in [12, 80]$ and random sampled CoT solutions with each step operators $t \in [1, 12]$. All other parameters are kept the same with the huggingface GPT-2 model. During the training process, the CoT indicator token `<t>` is also trained, so that during test-time, we can let the model decide which type of CoT it will use by only prompting the model with the question. For each model, we train 25000 iterations with batch size that equals 256. During test-time, we test 100 questions for each $T$ and $t$. All experiments can be conducted on one NVIDIA A800 80G GPU.

## H.2 Observation of subtask loss

As we observed in training losses, the loss of subtask generation tokens (e.g. $1 + 2$) for the easiest subtask($t = 1$) is about 3 times larger than the hardest subtask ($t = 12$), while the loss ratio for subtask answer tokens is $1e4$. Therefore, it is acceptable for taking the subtask error rate constant with $t$.

Besides, there is no obvious pattern showing the model sizes affect the subtask loss. Moreover, the smallest model and the largest model have almost the same subtask loss. Therefore, in our settings, we take model size as irrelevant with the subtask error rate.

## I Theoretical Results under Broader Scenarios

### I.1 General Errors

In the simple case we discussed in Section 4, we discussed the trend of overall accuracy with respect to $N$ and the variation of optimal $N$ with $M$ and $T$, assuming the subtask error rate is a linear function. In the following discussion, we aim to derive conclusions corresponding to more general error rate functions. We find that as long as the error function satisfies some basic assumptions on the **monotonicity** and **convexity** of the error functions, the above conclusions still hold.

**Assumption I.1.** $E(N, M, T)$ satisfies the following reasonable conditions:

- $0 < E(N = 1, M, T) < 1$

- $\lim_{N \to +\infty} E(N, M, T) = 0$

- $E(N, M, T)$ is monotonically deceasing with $N$, since more detailed decomposition leads to easier subtask.

- $E(N, M, T)$ is convex with $N$, since the benefits of further decomposing an already fine-grained problem($N$ is large) are less than the benefits of decomposing a problem that has not yet been fully broken down($N$ is small).

- $E(N, M, T)$ is monotonically deceasing with $M$, since stronger models have less subtask error rate.

- $E(N, M, T)$ is monotonically increasing with $T$, since harder total task leads to harder subtask while $N, M$ are the same.

**Assumption I.2.** $\sigma(T)$ is monotonically increasing with $T$

With Assumption I.1 and I.2), the core insights from the linear case can be generalized.

**Theorem I.3.** *For a noise function* $0 < \sigma(T) < 1$ *and a subtask error rate function* $0 < E(N, M, T) < 1$ *satisfying Assumptions I.1 and I.2, the general final accuracy function* $A(N)$ *from Proposition 4.1 has the following properties:*

- $\lim_{N \to +\infty} A(N) = 0$. *(Excessively long chains always fail.)*

- *If* $A(N)$ *has a maximum at* $N^* > 1$*, then* $N^*$ *has a lower bound related to* $M$ *and* $T$:

$$N^* \geq N_{LB}(M, T) = E_N^{-1}\left(1 - \frac{1}{e^2(1 - \sigma(T))}; M, T\right), \tag{6}$$

*where* $E_N^{-1}(\cdot; M, T)$ *is the inverse of* $E(N, M, T)$ *with respect to* $N$.

The monotonicity of $E_N^{-1}$ with respect to $M$ (decreasing) and $T$ (increasing, assuming $\sigma(T)$ doesn't dominate adversely) implies that the qualitative scaling laws (Corollaries stemming from Theorem 4.2) still hold under general conditions, supporting the empirically observed Simplicity Bias and the inverted U-shaped performance.

**Corollary I.4.** *As the model becomes stronger,* $E^{-1}$ *decreases monotonically with respect to* $M$, *which leads to a decrease of* $N(M, T)$.

**Corollary I.5.** *As the task becomes harder,* $E^{-1}$ *is monotonically increasing with respect to* $T$, *which leads to an increase in* $N(M, T)$.

## I.2 RANDOM ERROR

In Theorem 4.2 and I.3, we make a strong assumption that all sub-question or sub-answer errors are identical, which does not align well with real-world scenarios. In practice, each sub-task may exhibit a different error rate. However, they generally follow a trade-off: the more the task is decomposed, the easier each sub-task becomes. Specifically, we can model the error rate of each sub-task as a random variable with a fixed expectation that monotonically decreases with the number of CoT steps $N$.

To simplify the problem, here we assume $\sigma_i \sim B(\alpha_1(T), \beta_1(T))$ to be the sub-question error rate, and $e_i \sim B(\alpha_2(N, M, T), \beta_2(N, M, T))$ to be the sub-answer error rate. Then, as a variant of Proposition 4.1, the expectation of final accuracy is $\mathbb{E}\left[\prod_{i=1}^{N}(1 - e_i)(1 - \sigma_i)\right]$.

It is worth noting that each $\sigma_i$ or $e_i$ is not independent. If most steps are easy (i.e., have low error rates), the remaining steps are more likely to be easy as well. Moreover, if a particular step serves as a self-validation step, its high accuracy can influence the correctness of other steps that depend on it. This also provides an interpretation for reasoning models exhibiting backtracking behavior.

**Theorem I.6.** *Let $\alpha_1 = T$, $\beta_1 = C - T$, $\alpha_2 = T$, and $\beta_2 = NM - T$. Then the expected error rates for sub-questions and sub-answers are given by $\mathbb{E}[\sigma_i] = \frac{T}{C}$ and $\mathbb{E}[e_i] = \frac{T}{MN}$, respectively. Based on these estimates, we can derive an upper bound $\hat{A}(N)$ on the final accuracy*

$$\mathbb{E}\left[\prod_{i=1}^{N}(1 - e_i)(1 - \sigma_i)\right] \le \hat{A}(N) = \left[\left(1 - \frac{T}{C + 2N - 1}\right)\left(1 - \frac{T}{NM + 2N - 1}\right)\right]^N,$$

*which initially increases and then decreases as the number of CoT steps $N$ grows.*

This suggests that even with stochasticity, the fundamental trade-off leading to an optimal CoT length persists.

## J PROOF

In this section, we provide the proofs for all theorems.

### J.1 PROOF OF PROPOSITION 4.1

**Proposition 4.1.** *Assuming stepwise stationarity and independence conditioned on history, the final accuracy takes the form*

$$A(N) = P(a_{final} = a_{final}^* \mid q, \theta, N) = \alpha\big((1 - \sigma(T))(1 - E(N, M, T))\big)^N, \tag{1}$$

*where $\alpha$ denotes a constant independent of $N$.*

*Proof.* In each subtask $t_i$, which contains $t$ operators, there are $2t + 1$ tokens (as the number of numerical tokens is one more than the number of operators). Therefore, the accuracy of each subtask is given by

$$P(t_i = t_i^* | H_{i-1}, q, \theta) = (1 - \sigma(T))^{2t+1}. \tag{7}$$

In our theoretical analysis, for simplicity, we allow $t$ to be a fraction, defined as $t = \frac{T}{N}$, and assume that each subtask has the same level of difficulty given $T$ and $N$. Under this assumption, we have the

final accuracy:

$$A(N) = P(a_N = a_N^*|q, \theta) \tag{8}$$

$$= \prod_{i=1}^{N} P(t_i = t_i^*|H_{i-1}, q, \theta) P(a_i = a_i^*|t_i, H_{i-1}, q, \theta) \tag{9}$$

$$= \prod_{i=1}^{N} (1 - \sigma(T))^{2t+1} (1 - E(N, M, T)) \tag{10}$$

$$= (1 - \sigma(T))^{N(2t+1)} (1 - E(N, M, T))^N \tag{11}$$

$$= (1 - \sigma(T))^{2T} ((1 - E(N, M, T))(1 - \sigma(T)))^N \tag{12}$$

$$= \alpha ((1 - E(N, M, T))(1 - \sigma(T)))^N \tag{13}$$

$$\square$$

### J.2 Proof of Theorem 4.2

**Theorem 4.2** (Optimal CoT length). *There exists an optimal $N^*(M, T)$ maximizing $A(N)$:*

$$N^*(M, T) = \frac{T Z}{M(Z+1)}, \quad Z = W_{-1}\left(-\left(1 - \frac{T}{Ce}\right)\right),$$

*where $W_{-1}$ is the negative branch of the Lambert W function ($we^w = x$).*

*Proof.* Given Eq. (1) that

$$A(N) = \alpha \left(\left(1 - \frac{T}{C}\right)\left(1 - \frac{T}{NM}\right)\right)^N \tag{14}$$

We consider function

$$f(x) = \left[\left(1 - \frac{T}{Mx}\right)\left(1 - \frac{T}{C}\right)\right]^x. \tag{15}$$

For convenience, define

$$g(x) = \ln(f(x)) = x \ln\left[\left(1 - \frac{T}{Mx}\right)\left(1 - \frac{T}{C}\right)\right].$$

Thus,

$$g'(x) = \left[\ln\left(1 - \frac{T}{Mx}\right) + \frac{T}{Mx\left(1 - \frac{T}{Mx}\right)}\right] + \ln\left(1 - \frac{T}{C}\right).$$

Set $g'(x) = 0$:

$$\ln\left[\left(1 - \frac{T}{Mx}\right)\left(1 - \frac{T}{C}\right)\right] + \frac{T}{Mx\left(1 - \frac{T}{Mx}\right)} = 0.$$

Let $A = \frac{1}{1 - \frac{T}{Mx}}$, then we have

$$\ln\left[\left(1 - \frac{T}{C}\right)\right] + A - 1 = \ln(A).$$

Let $z := 1 - T/C$. (Since $T/C < 1$, $z = 1 - T/C > 0$.) By moving terms, we have:

$$-\frac{z}{e} = -A\exp(-A).$$

Therefore,

$$A = -W^{-1}\left(-\frac{z}{e}\right) = -Z,$$

Finally, we have

$$N(M, T) = x = \frac{TZ}{M(Z+1)}$$

Here $W(\cdot)$ is the **Lambert W function**, and for $0 < 1 - \frac{T}{C} < 1$, the argument $\alpha = -\frac{1-T/C}{e}$ lies in the interval $\left(-\frac{1}{e}, 0\right)$. This means there are two real branches $W_0$ and $W_{-1}$ in that domain, but since $\frac{Z}{Z+1} > 0$, we have $Z < -1$. Therefore, we only take the solution on branch $W_{-1}$. $\square$

### J.3 PROOF OF COROLLARY 4.3

**Corollary 4.3** (Scaling laws). *From Theorem 4.2:*

- *$N^*(M, T)$ increases with $T$ (harder tasks warrant longer CoT).*

- *The optimal operators per step $t^* = T/N^*(M, T) = M(1 + 1/Z)$ increases with $T$ (envelope behavior).*

- *$N^*(M, T)$ decreases with $M$ (stronger models need fewer steps).*

*Proof.* The second and third conclusions can be easily derived through monotonic composition, so we primarily focus on proving the first point. We begin the proof by incorporating the notation from J.2. We have

$$g'(x) = \left[\ln\left(1 - \frac{T}{Mx}\right) + \frac{T}{Mx\left(1 - \frac{T}{Mx}\right)}\right] + \ln\left(1 - \frac{T}{C}\right),$$

and $x^*(T)$ such that $g'(x^*(T)) = 0$.

Let $F(x^*(T), T) = g'(x^*(T)) = 0$ We want to see how $x^*(T)$ changes as $T$ changes, therefore we take total derivative w.r.t. $T$. By the chain rule,

$$0 = \frac{d}{dT} F(x^*(T), T) = \underbrace{\frac{\partial F}{\partial x}(x^*(T), T)}_{\text{call this } F_x} \cdot \frac{\partial x^*}{\partial T}(T) + \underbrace{\frac{\partial F}{\partial T}(x^*(T), T)}_{\text{call this } F_T}.$$

Hence

$$\frac{\partial x^*}{\partial T}(T) = -\frac{F_T\left(x^*(T), T\right)}{F_x\left(x^*(T), T\right)}.$$

So the sign of $x'^*(T)$ is the opposite of the sign of $F_T$, provided $F_x \neq 0$.

Since

$$F_x\left(x, T\right) = -\frac{T^2}{x(Mx - T)^2} < 0, \forall x > 0, \tag{16}$$

all we need to prove is

$$F_T\left(x^*(T), T\right) = \frac{T}{(Mx^*(T) - T)^2} - \frac{1}{C - T} > 0. \tag{17}$$

That is

$$\frac{\sqrt{T(C - T)} + T}{M} > x^*(T). \tag{18}$$

Let $x_0(T) = \frac{\sqrt{T(C-T)}+T}{M}$ be the test point.

According to Lemma J.1, $F(x_0(T), T) < 0$. Since $F(x^*(T), T) = 0$, and $F_x\left(x^*(T), T\right) < 0$, we have $x_0(T) > x^*(T)$.

Thus, $F_T\left(x^*(T), T\right) > 0$ holds and we have proved our corollary with $\frac{\partial x^*}{\partial T}(T) > 0$.

$\square$

## J.4 Proof of Theorem I.3

**Theorem I.3.** *For a noise function $0 < \sigma(T) < 1$ and a subtask error rate function $0 < E(N, M, T) < 1$ satisfying Assumptions I.1 and I.2, the general final accuracy function $A(N)$ from Proposition 4.1 has the following properties:*

- $\lim_{N \to +\infty} A(N) = 0$. *(Excessively long chains always fail.)*

- *If $A(N)$ has a maximum at $N^* > 1$, then $N^*$ has a lower bound related to $M$ and $T$:*

$$N^* \geq N_{LB}(M, T) = E_N^{-1}\left(1 - \frac{1}{e^2(1 - \sigma(T))}; M, T\right), \tag{6}$$

*where $E_N^{-1}(\cdot; M, T)$ is the inverse of $E(N, M, T)$ with respect to $N$.*

*Proof.* (1) Since $0 < A(N) < (1 - \sigma(T))^N$, and $\lim_{N \to +\infty}(1 - \sigma(T))^N = 0$, $\lim_{N \to +\infty} A(N, M, T) = 0$

(2) Let $g(x)$ denote $E(x, M, T)$ and define $f(x) = \ln A(x)$. Then,

$$f'(x) = \ln(1 - \sigma(T)(1 - g(x))) - \frac{xE'(x)}{1 - E(x)} \tag{19}$$

$$< \ln(1 - \sigma(T)(1 - g(x))) + 2, \quad \text{(since $E$ is convex and $x = N \geq 1$)} \tag{20}$$

If $A(N)$ attains its maximum at some point $N^* > 1$, then $\ln(1 - \sigma(T)) + 2 > 0$. Otherwise, we would have $f'(x) < \ln(1 - \sigma(T)) + 2 \leq 0 \ \forall x > 1$, leading to a contradiction.

Thus, it follows that $e^2(1 - \sigma(T)) > 1$.

Now, define $N(M, T) = E^{-1}\left(1 - \frac{1}{e^2(1-\sigma(T))}\right)$, which satisfies

$$\ln(1 - \sigma(T)(1 - g(N(M, T)))) + 2 = 0.$$

If there exists $x^* < N(M, T)$ such that $f'(x^*) = 0$, then we obtain

$$0 = f'(x^*) < \ln(1 - \sigma(T)(1 - E(x))) + 2 < 0,$$

which is a contradiction. Hence, the assumption that $x^* < N(M, T)$ must be false.

Therefore, we conclude that $x^* = N^* > N(M, T)$.

$\square$

## J.5 Proof of Theorem I.6

**Theorem I.6.** *Let $\alpha_1 = T$, $\beta_1 = C - T$, $\alpha_2 = T$, and $\beta_2 = NM - T$. Then the expected error rates for sub-questions and sub-answers are given by $\mathbb{E}[\sigma_i] = \frac{T}{C}$ and $\mathbb{E}[e_i] = \frac{T}{MN}$, respectively. Based on these estimates, we can derive an upper bound $\hat{A}(N)$ on the final accuracy*

$$\mathbb{E}\left[\prod_{i=1}^{N}(1 - e_i)(1 - \sigma_i)\right] \leq \hat{A}(N) = \left[\left(1 - \frac{T}{C + 2N - 1}\right)\left(1 - \frac{T}{NM + 2N - 1}\right)\right]^N,$$

*which initially increases and then decreases as the number of CoT steps $N$ grows.*

*Proof.* According to the multidimensional version of Hölder's inequality,

$$\mathbb{E}\left[\prod_{i=1}^{N}(1-e_i)(1-\sigma_i)\right] \le \prod_{i=1}^{N}\left(\mathbb{E}[(1-e_i)^{2N}]\mathbb{E}[(1-\sigma_i)^{2N}]\right)^{\frac{1}{2N}} \tag{21}$$

$$\text{(Lemma J.2)} \le \prod_{i=1}^{N}\left(1-\frac{T}{C+2N-1}\right)\left(1-\frac{T}{NM+2N-1}\right) \tag{22}$$

$$= \left[\left(1-\frac{T}{C+2N-1}\right)\left(1-\frac{T}{NM+2N-1}\right)\right]^{N} \tag{23}$$

$\square$

### J.6  PROOF OF COROLLARY 4.4

**Corollary 4.4** (RL Converges to Optimal CoT Length). *For gradient ascent on $J(\theta)$ with sufficiently small step size, the policy converges to a deterministic solution $\pi_\theta(N_i) = 1$ iff $i = \arg\max_j A(N_j)$. Thus, RL training converges to the optimal CoT length $N^* = \arg\max_{N\in\mathcal{A}} A(N)$.*

*Proof.* We treat the choice of CoT length as a $k$-armed stochastic bandit with action set $\mathcal{A} = \{N_1, \ldots, N_k\}$ and unknown success probabilities[2] $A(N_i) \in (0,1)$. Without loss of generality, relabel the arms so that

$$A(N_1) = \max_j A(N_j) =: A^*, \qquad A(N_1) \ge A(N_2) \ge \cdots \ge A(N_k).$$

The agent uses a softmax (Gibbs) policy

$$\pi_\theta(N_i) = \frac{e^{\theta_i}}{\sum_{j=1}^{k} e^{\theta_j}}, \qquad \theta \in \mathbb{R}^k, \tag{24}$$

and maximises the expected reward

$$J(\theta) = \sum_{i=1}^{k} \pi_\theta(N_i)\, A(N_i). \tag{25}$$

Because $\pi_\theta$ is $C^\infty$ in $\theta$ and $A(N_i)$ are constants, $J$ is smooth.

Under the REINFORCE estimator with sufficiently small, fixed step size $\eta > 0$, gradient ascent updates take the form

$$\theta^{(t+1)} = \theta^{(t)} + \eta\,\nabla_\theta J\big(\theta^{(t)}\big), \tag{26}$$

where

$$\frac{\partial J}{\partial \theta_i} = \pi_\theta(N_i)\Big(A(N_i) - J(\theta)\Big). \tag{27}$$

Eq. (27) is the classical *replicator* (or logit) gradient. Define the simplex $\Delta^{k-1} := \{\pi \in (0,1]^k \mid \sum_i \pi_i = 1\}$ and write $\pi_\theta = (\pi_\theta(N_1), \ldots, \pi_\theta(N_k))$.

Letting $\eta \to 0$ yields the ODE

$$\dot{\pi}_i = \pi_i\big(A(N_i) - \langle \pi, A \rangle\big), \qquad i = 1, \ldots, k, \tag{28}$$

with $\langle \pi, A \rangle = \sum_j \pi_j A(N_j)$. Eq. (28) is the **replicator dynamics** for a fitness landscape $A$ on $\Delta^{k-1}$.

Consider the Kullback–Leibler divergence to the optimal pure strategy $\mathbf{e}_1 = (1, 0, \ldots, 0)$,

$$V(\pi) = \sum_{i=1}^{k} \pi_i \ln\Big(\frac{\pi_i}{e_{1,i}}\Big) = -\ln \pi_1.$$

---

[2] By Proposition 4.1, $A(N_i)$ is the probability that the final answer is correct when a chain of length $N_i$ is used. The bandit is *stationary* because $A(N_i)$ does not depend on time or the agent's past actions.

$V$ is non-negative on $\Delta^{k-1}$ and $V(\pi) = 0$ iff $\pi = \mathbf{e}_1$.

Taking the time derivative along Eq. (28) gives

$$\frac{dV}{dt} = -\frac{\dot{\pi}_1}{\pi_1} = -\big(A(N_1) - \langle \pi, A \rangle\big) \leq 0,$$

with equality iff $\pi_1 = 1$ *or* $A(N_1) = \langle \pi, A \rangle$. The latter can only happen if $\pi_1 = 1$ because $A(N_1) > A(N_j)$ for $j > 1$. Hence $V$ is a strict Lyapunov function, and $\mathbf{e}_1$ is the *unique* asymptotically stable equilibrium of Eq. (28). All other stationary points (mixtures over sub-optimal arms) are unstable.

For sufficiently small but fixed $\eta$ (choose $\eta < \frac{1}{A^*}$, which always exists), projected gradient ascent is a *perturbed* discretisation of Eq. (28). Standard results for primal-space mirror descent imply that the discrete iterates $\pi^{(t)} \equiv \pi_{\theta(t)}$ converge almost surely to the set of asymptotically stable equilibria of the ODE, i.e. to $\{\mathbf{e}_1\}$. Therefore

$$\lim_{t \to \infty} \pi_{\theta(t)}(N_i) = \begin{cases} 1, & \text{if } i = \arg\max_j A(N_j), \\ 0, & \text{otherwise.} \end{cases}$$

Because $A$ may attain its maximum at several arms, the limit is a deterministic policy that places all probability on *some* maximiser of $A$.

Thus gradient ascent on Eq. (25) converges to a deterministic policy that always selects an optimal CoT length $N^* = \arg\max_{N \in \mathcal{A}} A(N)$, completing the proof. $\square$

### J.7 Technical Lemmas

**Lemma J.1** (test point). *Let $F(x)$ be defined as*

$$F(x) = \ln\left(1 - \frac{T}{Mx}\right) + \frac{T}{Mx\left(1 - \frac{T}{Mx}\right)} + \ln\left(1 - \frac{T}{C}\right),$$

*where $T, M, C \in \mathbb{R}^+$ satisfy the conditions:*

- $0 < \frac{T}{C} < 0.9$,

- $0 < \frac{T}{Mx} < 1$.

*Define $x_0$ as*

$$x_0 = \frac{\sqrt{T(C - T)} + T}{M}.$$

*Then, we have*

$$F(x_0) < 0.$$

*Proof.* At $x = x_0$, note that

$$Mx_0 = \sqrt{T(C - T)} + T.$$

Thus,

$$1 - \frac{T}{Mx_0} = 1 - \frac{T}{T + \sqrt{T(C - T)}} = \frac{\sqrt{T(C - T)}}{T + \sqrt{T(C - T)}}.$$

Therefore,

$$\ln\left(1 - \frac{T}{Mx_0}\right) = \ln\left(\frac{\sqrt{T(C - T)}}{T + \sqrt{T(C - T)}}\right) = \ln\sqrt{T(C - T)} - \ln\left(T + \sqrt{T(C - T)}\right).$$

Also, observe that

$$\frac{T}{Mx_0\left(1 - \frac{T}{Mx_0}\right)} = \frac{T}{(T + \sqrt{T(C - T)})\left(\frac{\sqrt{T(C-T)}}{T+\sqrt{T(C-T)}}\right)} = \frac{T}{\sqrt{T(C - T)}} = \sqrt{\frac{T}{C - T}}.$$

It is convenient to introduce the change of variable

$$u = \sqrt{\frac{T}{C - T}},$$

so that

$$T = u^2(C - T), \quad \sqrt{T(C - T)} = u(C - T).$$

Then we have

$$T + \sqrt{T(C - T)} = u^2(C - T) + u(C - T) = u(C - T)(u + 1).$$

In these terms we have:

$$\ln \sqrt{T(C - T)} = \ln\left[u(C - T)\right] = \ln u + \ln(C - T),$$

$$\ln\left(T + \sqrt{T(C - T)}\right) = \ln\left[u(C - T)(u + 1)\right] = \ln u + \ln(C - T) + \ln(u + 1),$$

and

$$\sqrt{\frac{T}{C - T}} = u.$$

Finally, we have

$$\ln\left(1 - \frac{T}{C}\right) = -\ln\left(\frac{C}{C - T}\right) = -\ln(u^2 + 1)$$

Thus, the function $F(x_0)$ becomes

$$F(x_0) = \ln u + \ln(C - T) - \left(\ln u + \ln(C - T) + \ln(u + 1)\right) + u - \ln(u^2 + 1) \quad (29)$$

$$= -\ln(u + 1) + u - \ln(u^2 + 1), \quad (30)$$

where $u = \sqrt{\frac{T}{C-T}} \in (0, 3)$. It is easy to show $F(x_0) < 0$ when $u \in (0, 3)$. $\qquad\square$

**Lemma J.2** (Estimation of the $n$-th Moment of the Beta Distribution). *Let $x \sim \mathrm{Beta}(\alpha, \beta)$. Then*

$$\mathbb{E}[(1 - x)^n] \leq \left(1 - \frac{\alpha}{\alpha + \beta + n - 1}\right)^n.$$

*Proof.*

$$\mathbb{E}[(1 - x)^n] = \frac{1}{B(\alpha, \beta)} \int_0^1 (1 - x)^n x^{\alpha-1}(1 - x)^{\beta-1} \, dx$$

$$= \frac{1}{B(\alpha, \beta)} \int_0^1 x^{\alpha-1}(1 - x)^{\beta+n-1} \, dx$$

$$= \frac{B(\alpha, \beta + n)}{B(\alpha, \beta)}$$

$$= \frac{\Gamma(\alpha)\Gamma(\beta + n)}{\Gamma(\alpha + \beta + n)} \cdot \frac{\Gamma(\alpha + \beta)}{\Gamma(\alpha)\Gamma(\beta)}$$

$$= \frac{\Gamma(\beta + n)}{\Gamma(\beta)} \cdot \frac{\Gamma(\alpha + \beta)}{\Gamma(\alpha + \beta + n)}$$

$$= \prod_{i=0}^{n-1} \frac{\beta + i}{\alpha + \beta + i}$$

$$\leq \left(\frac{\beta + n - 1}{\alpha + \beta + n - 1}\right)^n$$

$$= \left(1 - \frac{\alpha}{\alpha + \beta + n - 1}\right)^n.$$

$$\square$$

## K  PSEUDO-CODE OF LENGTH-FILTERED VOTE

---

**Algorithm 1** Length-filtered Vote

---

1: **Input:** Model $f_\theta$, Question $q$, Space of All Possible Answers $A$, Number of Total Groups $M$, Number of Selected Groups $K$, Group Width $D$
2: **Output:** Final Answer $\hat{a}$
3: Sample candidates $c_1, \ldots, c_n \overset{i.i.d.}{\sim} f_\theta(q)$
4: **Define** $\mathcal{A}(c)$ as the corresponding answer of candidates $c$.
5: **Define** $p_j \in [0,1]^{|\mathcal{A}|}$ as the frequency of each answer in length group $L_j$.
6: **for** $j = 1$ to $m$ **do**
$\qquad L_j = \{c_i \mid \ell(c_i) \in [D * (j-1), D * j), i = 1, \cdots, n\}$
7:     **for** $a \in \mathcal{A}$ **do**
$$p_j[a] = \frac{\sum_{c \in L_j} \mathbb{I}(\mathcal{A}(c) = a)}{|L_j|}$$
8:     **end for**
9: **end for**
10: $\{s_1, \ldots, s_K\} = \arg\min_{S \subseteq \{1,\ldots,M\}, |S|=K} \sum_{s \in S} H(p_s)$
11: $\hat{a} = \arg\max_{a \in A} \sum_{c \in L_{s_1} \cup \cdots \cup L_{s_K}} \mathbb{I}(\mathcal{A}(c) = a)$
12: **return** $\hat{a}$

---

