# OpenReview forum: "When More is Less: Understanding Chain-of-Thought Length in LLMs"
_ICLR.cc/2026/Conference — ICLR 2026 Poster_

### Official Review · Reviewer_pd7o · 2025-10-28

**Soundness:** 3
**Presentation:** 3
**Contribution:** 3
**Rating:** 6
**Confidence:** 4

**Summary:**

The paper studies how CoT length affects accuracy and cost. It reports an inverted U-shaped relationship between CoT length and task accuracy: performance improves up to an optimal CoT length, then degrades due to error accumulation; the optimal length increases with task difficulty and decreases with model capability. The authors present (i) evidence on real LLMs (Qwen2.5 series) and synthetic setups (arithmetic, DP), (ii) an error-accumulation theory that yields a closed-form for the optimal length and scaling laws, and (iii) two applications: training with optimal-length CoTs and an inference-time Length-Filtered Vote that selects answers from near-optimal-length samples. They also show that RL can calibrate CoT length toward the optimum.

**Strengths:**

This paper studies an important problem about CoT length and the empirical analysis provides clear patterns: Multiple figures show the inverted-U accuracy vs. length and the stated scaling behaviors (harder tasks → longer optimal length; larger models → shorter optimal length). The well-controlled synthesis experiments are insightful.

The theoretical study strengthens the insight of this work: A compact error-accumulation model derives an explicit optimal length and recovers the empirical scaling laws.

The provided practical implication provides actionable takeaways and can inspire future study.

Overall, this is a comprehensive and insightful paper combining both analysis and practical usage.

**Weaknesses:**

1. Missing experimental details, especially in Section 2.

a) The definition of CoT length is very vague, and the authors do not provide details of how to extract the length, such as whether use another model or parsing based on some rules.

b) The definition of task difficulty also requires proper thought as the accuracy can be biased. For example, if the model favors more CoT steps on one harder question (the difficulty from a human's perspective) and achieves higher accuracy, while generating shorter steps for another simpler question and achieving lower accuracy. This can lead to the wrong problem difficulty. More proper ways include leveraging the dataset's own difficulty level, human judgment, advanced LLM's judgment and etc.

2. There exists a concern about whether the optimal CoT length phenomenon on synthetic data can scale up to larger models.

a) The Arithmetic Problem seems to be too simple for advanced mdoels (maybe even for small-size models in the Qwen2.5 family), and I suspect that advanced models can solve these questions in one step and therefore make this dataset trivial. I suggest that the author extend the Arithmetic Problem from only addition to include more complex operators.

b) The DP problem is interesting, and it is worth testing on more advanced models.

c) The claim of per-step difficulty can benefit from more experiments. There may also exist an optimal per-step difficulty as the model can not handle sub-tasks with infinitely increasing difficulty (otherwise, one step is already optimal). This is crucial as it can guide how to define sub-tasks based on the difficulty.

3. While the theoretical analysis is intuitive, it relies on strong assumptions that may only hold for the Arithmetic Problem. This is not a very critical problem, as I understand the difficulty in modeling CoT errors. However, the authors should explicitly mention the assumptions in the main paper; otherwise, people may be confused about why $\sigma(T)$ and $E(N,M,T)$ are independent of the step index, and the sub-question error only relies on the total difficulty (in proposition 4.1). This concern arises because in practice, steps are generated autoregressively and later step depends on previous steps. Please make these clear.

**Questions:**

See weakness

---

> ### Author Response · Authors · 2025-11-21
>
> We thank the reviewer pd7o ****for the positive and encouraging evaluation. We appreciate the recognition of our empirical findings and the insights they reveal. We are also grateful for the acknowledgment of our theoretical error-accumulation analysis, as well as the practical implications and actionable takeaways enabled by our work. We address your concerns below.
>
> ---
>
> **W1:** Missing experimental details, especially in Section 2.
>
> - a) The definition of CoT length is very vague
>
>     **A1.a:** As noted at the beginning of Sec. 2 (line 138-139), *CoT length is measured by the number of intermediate reasoning steps* and *the full details of step segmentation and length control are provided in Appendix D.* In details, we follow the common practice [1] and segment steps using the newline delimiter “\n”. In the new version of pdf, we have made this definition more clear.
>
>
> - b) The definition of task difficulty also requires proper thought as the accuracy can be biased.
>
>     **A1.b:** We agree that task difficulty requires careful definition because accuracy can be biased by a model’s intrinsic preference for certain CoT lengths. We address this concern in two ways:
>
>     - As you noted, different CoT-length preferences across difficulty levels could accidentally make “easier” questions appear harder. To mitigate this, when generating the 60 responses per question, we explicitly prompt the model to produce *multiple* CoT lengths (see Appendix D.1, Solution Length Control). This reduces the impact of the model’s internal length bias on the measured accuracy.
>     - Task difficulty is inherently *model-dependent*: a question that is easy for one model may be hard for another due to differences in training data and capabilities. Thus, we define difficulty in a model-aware manner by using the *average accuracy* of the target model as the indicator. This ensures that difficulty reflects the model’s actual competence rather than an externally imposed notion.
>
>     We have added this discussion into Appendix D.4.
>
>     [1] Complexity-Based Prompting for Multi-Step Reasoning. Fu et al. ICLR 2023
>
> ---
> **W2:** There exists a concern about whether the optimal CoT length phenomenon on synthetic data can scale up to larger models.
>
> - a) The Arithmetic Problem seems to be too simple for advanced models (maybe even for small-size models in the Qwen2.5 family)
>
>     **A2.a:** Thank you for raising this concern. We respectfully disagree with the assumption that the Arithmetic Problem is “too simple” for advanced models. For instance, on a **24-operator arithmetic task**, **Qwen3-Max (w/ or w/o Thinking mode)** **fails** consistently when tool use is disabled. This indicates that performance degrades substantially with increasing complexity, and that **the task remains non-trivial even for state-of-the-art models**.
>
>     More importantly, the contribution of our synthetic dataset does not lie in the arithmetic operation itself, but in its **systematic construction of multiple CoT solutions with different lengths and granularities for the same underlying problem**. To our knowledge, this is the first synthetic dataset that provides *aligned multi-length CoTs*, enabling controlled studies of reasoning length that prior datasets do not support.
>
>
> - **b)** The DP problem is interesting, and it is worth testing on more advanced models.
>
>     **A2.b:** Thank you for the suggestion. We evaluated the Qwen2.5 instruct models on our dynamic-programming (DP) dataset with depths 6, 8, and 10. Because the problem difficulty is precisely controllable, we tested 100 instances per difficulty level and sampled 10 CoT generations per instance. Following our real-world evaluation protocol, we measured the **optimal CoT length** for each model–task pair.
>
>     While the 1.5B model achieved <10% accuracy and was therefore excluded from analysis. Notably, even the 72B model does **not** achieve perfect accuracy on depth-6 problems (81.8%), despite these appearing straightforward to humans. This confirms that **our** **DP dataset is non-trivial** and effectively probes algorithmic reasoning.
>
>     As shown in Table 1, larger models consistently use shorter optimal CoT lengths, while harder problems require longer optimal lengths, which both consistent with our theories and previous experimental results. We have added this experiment in Appendix C.
>
>     Table 1. Optimal CoT length (accuracy %) across different model sizes and DP task depth
>
>     | **DP Depth →** | **6** | **8** | **10** |
>     | --- | --- | --- | --- |
>     | **7B** | 5 (35.0%) | 7 (30.3%) | 8 (22.6%) |
>     | **32B** | 4 (70.6%) | 5 (45.8%) | 8 (42.9%) |
>     | **72B** | 3 (**81.8%**) | 5 (66.7%) | 6 (46.2%) |

---

> ### Author Response · Authors · 2025-11-21
>
> - **c)** The claim of per-step difficulty can benefit from more experiments. There may also exist an optimal per-step difficulty as the model can not handle sub-tasks with infinitely increasing difficulty (otherwise, one step is already optimal). This is crucial as it can guide how to define sub-tasks based on the difficulty.
>
>     **A2.c:** We agree that understanding the optimal per-step difficulty is important, and we provide both theoretical and empirical evidence supporting this phenomenon. As shown in **Figure 3(a)** and formalized in **Corollary 4.3**, the *optimal per-step difficulty* indeed **increases** as the total task difficulty increases. However, this does **not** imply that *one step should already be optimal.* The key reason is that the optimal per-step difficulty grows slower than total task difficulty.
>
>     Overall, this leads to a structurally interesting conclusion: As task difficulty increases, both the *optimal number of sub-questions*  and *optimal difficulty of each sub-question* increase*.* This suggests that **models must learn to improve both the complexity they can handle per step and the number of reasoning steps they can reliably execute in order to solve harder problems**.
>
> ---
>
> **W3:** Why $\sigma(T)$ and $E(N,M,T)$  are independent of the step index, and the sub-question error only relies on the total difficulty (in proposition 4.1). This concern arises because in practice, steps are generated autoregressively and later step depends on previous steps. Please make these clear.
>
> **A3:** Thank you for raising this point. We agree that the assumptions underlying our theoretical model should be stated more explicitly in the main paper, as they are essential for interpreting Proposition 4.1.
>
> - Why $\sigma(T)$ and $E(N,M,T)$  are independent of the step index?
>
>     Theorem 4.2 analyzes the simplest deterministic case for clarity, where we treat each error rate as independent scaler. However, as noted in the last paragraph of Section 4 (line 371, “Extension to Nonlinear and Stochastic Error Functions”), our framework already extends to a more realistic setting. In this version:
>
>     - Each sub-question error is modeled as a random variable following a Beta distribution with a shared expectation.
>     - Sub-question errors are **correlated.** It considers when the total CoT length is fixed, easier sub-questions imply harder sub-questions elsewhere.
>     - The same holds for sub-answer errors.
>
>     Under this stochastic formulation, we still obtain that the **expected final accuracy is upper-bounded by an inverted-U-shaped function** of the CoT length N. This aligns with and generalizes the intuition demonstrated in the simple case.
>
> - Why the sub-question error only relies on the total difficulty?
>
>     Our analysis focuses on the regime where the model has **already acquired the ability to produce CoTs of different lengths**, and the remaining source of errors comes from solving the sub-questions rather than from failing to format or decompose the solution.
>
>     This assumption is empirically justified, and we have provided the supporting evidence in Appendix H.2. Specifically, during training of our synthetic models, we examined the loss of sub-question tokens and sub-answer tokens across different difficulty levels.
>
>     - For sub-question tokens (e.g., “1 + 2”), the loss for the easiest tasks (t = 1) is roughly **3× larger** than for the hardest tasks (t = 12).
>     - For sub-answer tokens, this ratio is about $10^4$.
>     - Importantly, these losses show **no systematic dependence on model size**—they remain nearly identical across all model scales.
>
>     Therefore, in our setting, we reasonably treat the sub-question and sub-answer error rates as **independent of the model size**, and driven primarily by the task difficulty $t$.
>
>
> ---
>
> Thank you again for the insightful comments. If you find it satisfactory, we respectfully hope that you can re-evaluate our work and raise your score. We are happy to address your further concerns.

---

### Official Review · Reviewer_seMZ · 2025-10-29

**Soundness:** 3
**Presentation:** 4
**Contribution:** 4
**Rating:** 8
**Confidence:** 4

**Summary:**

This paper addresses the question of how long chain of thought (CoT) reasoning should be to achieve optimal performance in LLMs. They find that the the ideal CoT length generally increases as tasks get harder, and generally decreases as models become larger. They develop synthetic experiments to support their hypothesis, and further demonstrate their findings on real-world tasks for large models.

**Strengths:**

* This paper is well-written.
* As mentioned, the paper provides both synthetic and real-world tasks where the findings are observed.
* The authors meaningfully point out that reasoning traces typically become longer (most prominently demonstrated by the Deepseek-R1 paper), yet they show evidence that in fact this does not always true (e.g. on Leetcode-2k).
* The authors provide one important application of their work, which is to propose a novel majority voting mechanism weighted by reasoning length, which outperforms standard majority voting.

Overall I think that this paper makes important contributions toward addressing the fundamental question of how long reasoning should be in LLMs, and has important implications for how to design LLMs with a specific reasoning budget.

**Weaknesses:**

My main concern is that I do not think the experimental setup adequately discusses how important reflection/backtracking is to the reasoning process, which I think the authors rightfully point out is present in "real-world CoTs" (Section 2, Appendix A.3). To frame it another way, the main question I am asking is: given that real-world models do self-correct, **how does self-correction/backtracking play a role in influencing the total length of the CoT**?
* To my understanding, none of the synthetic setups (addition/dynamic programming) include some variant of verification/self-correction. The conclusion from the synthetic experiments seem to suggest more that  "LLMs can perform multiple calculations in one step", rather than "LLMs _require_ only X steps to solve the full problem".
* I wonder if it would be possible to augment the synthetic setup with some "mistake" in the reasoning process. For example in the addition case, one could artificially inject a wrong addition step (1+2+3+4=_11_), then follow-up with some self-reflection token in the next step and provide the corrected step. I think it would be interesting to see how much noise you add to the trace influences the final optimal length.
* I find Corollary 4.4 to be a little weak. It simply states that RL training optimizes for reward $A(N)$, which should be straightforward to see given that this is just the REINFORCE objective. The empirical analyses (Figure 2c, Fig 4) seem to suggest that CoT length keeps going down - I am wondering if the theory would say anything further about when CoT length goes up or down?

**Questions:**

* I am missing some intuition on why RL is causing CoT length to go down (Figure 2c). What is happening to the model rollouts? Is it learning to merge steps, or do less backtracking, etc?
* Similar question for Figure 4: what happens as the model goes from CoT length 24->5 - is it combining lots of steps together?
* Figure 1b: for task difficulty 24, seems to only get better with more CoT length. does this level off?

---

> ### Author Response · Authors · 2025-11-21
>
> We appreciate the reviewer seMZ’s recognition of our comprehensive experimental study, the counter-intuitive findings, and the practical insights and methods enabled by our analysis. We address your concerns below.
>
> ---
>
> **Q1:** I wonder if it would be possible to augment the synthetic setup with some "mistake" in the reasoning process.
>
> **A1:** Thank you for the thoughtful suggestion. Following your advice, **we augmented our synthetic arithmetic dataset to explicitly teach the model self-correction.** Concretely, we modified the training sequences so that after a reasoning step, the model occasionally encounters an intentionally incorrect intermediate result, immediately followed by a self-correction step that repeats the same reasoning with the correct value. Formally, we replace
>
> - question + **step₁ + ans**₁ + step₂ + …
>
> with
>
> - question + **\[step₁ + wrong_ans₁\](optionally) + step₁ + correct_ans**₁ + step₂ + …
>
> We control the proportion of injected incorrect steps by a parameter $p$, and through preliminary experiments, we found $p = 0.3$ provides the best balance between maintaining core computational ability and encouraging effective self-correction. All other training configurations remain unchanged, and we trained a 6-layered GPT2 model under this setting.
>
> At test time, we evaluate the model on tasks of fixed difficulty under different CoT lengths and select the optimal length as the one achieving the highest accuracy. When a model performs self-correction within a single step, **we count the original erroneous step together with its immediate correction as one CoT step**, since our notion of CoT length reflects how finely the problem is decomposed rather than how many times a local computation is revised.
>
> As shown in Table 1, models trained with self-correction converge to substantially shorter optimal CoT lengths (first two rows). Although this may seem counter-intuitive, it is fully consistent with our broader findings: **strengthening the model’s ability to repair local errors enables it to handle more difficult subtasks per step (last two rows), which in turn reduces the total CoT length required to reach maximal accuracy.**
>
> **Table 1. Optimal CoT length and subtask difficulty w/ and w/o self-correction (SC)**
>
> | **Task Difficulty** | **16** | **24** | **32** | **40** |
> | --- | --- | --- | --- | --- |
> | Optimal CoT length w/o SC | 4 | 5 | 8 | 10 |
> | Optimal CoT length w/ SC | 2 | 2 | 3 | 5 |
> | Optimal subtask difficulty w/o SC | 4 | 5 | 4 | 4 |
> | Optimal subtask difficulty w/ SC | 8 | 12 | 11 | 8 |
>
> ---
>
> **Q2:** How does self-correction/backtracking play a role in influencing the total length of the CoT?
>
> **A2:** Thank you for the question. In our framework, we distinguish *self-correction* from *backtracking*.
>
> By *self-correction*, we refer to cases where the model detects an error in its previous intermediate result and immediately rectifies it. As shown in A1, this ability improves the model’s capacity to solve more difficult subtasks within a single step, and therefore leads to a shorter optimal CoT length.
>
> *Backtracking*, in contrast, describes a broader search behavior in which the model explores multiple intermediate paths or sub-tasks before arriving at the final answer. We do not attempt to model specific cognitive processes as separate mechanisms. Instead, we treat all such behaviors as different forms of task decomposition. Regardless of whether the model proceeds linearly, corrects itself, or explores alternative paths, these are simply different ways of breaking the original problem into subtasks.
>
> Under this unified view, the CoT length is determined solely by the total number of resulting subtasks, and is intentionally agnostic to the internal reasoning style that produced them.
>
> ---
>
> **Q3:** The conclusion from the synthetic experiments seem to suggest more that "LLMs can perform multiple calculations in one step", rather than "LLMs *require* only X steps to solve the full problem".
>
> **A3:** We view these two statements as equivalent under our task-decomposition formulation. In this framework, “performing multiple calculations in one step” directly corresponds to “requiring fewer total steps to solve the full problem” when total difficulty is given.
>
> Empirically, this is exactly what Fig. 1(a,b) shows: accuracy first increases and then decreases as CoT length grows, implying the existence of an **optimal number of steps $N^*$**. This means that **LLMs *require* only $N^*$ steps to solve the problem most effectively**. Thus, the observation that LLMs can combine multiple operations in a single step is simply the mechanism that determines the optimal CoT length.

---

> ### Author Response · Authors · 2025-11-21
>
> **Q4:** I find Corollary 4.4 to be a little weak. It simply states that RL training optimizes for reward $A(N)$, which should be straightforward to see given that this is just the REINFORCE objective. The empirical analyses (Figure 2c, Fig 4) seem to suggest that CoT length keeps going down - I am wondering if the theory would say anything further about when CoT length goes up or down?
>
> **A4:** Thank you for the thoughtful question. In short, **our theory does specify when RL will increase or decrease CoT length**: RL will adjust the policy toward the CoT length that yields the highest expected accuracy. If the model’s current expected CoT length is *below* the optimal length, RL will push it upward; if it is *above* the optimal length, RL will push it downward.
>
> Corollary 4.4 formalizes this observation by treating different CoT lengths as distinct actions and showing that, under the REINFORCE objective, the policy concentrates probability mass on the accuracy-optimal length.
>
> To support this, our real-world and synthetic experiments demonstrate that different lengths indeed achieve different accuracies. Prior work[1,2] often reports that RL increases CoT length because base models tend to produce chains that are *shorter* than optimal. Our findings reveal the complementary case: when a model initially prefers overly long CoTs (e.g., due to training on long-chain datasets), RL calibrates the length downward, converging toward the accuracy-optimal value.
>
> Overall, both the theory and experiments indicate that RL does not inherently favor longer or shorter CoTs; it adapts the model toward the task-dependent optimal length.
>
> **Reference:**
>
> [1] Light-R1: Curriculum SFT, DPO and RL for Long COT from Scratch and Beyond, Wen et al, 2025
>
> [2] DeepSeek-R1: Incentivizing Reasoning Capability in LLMs via Reinforcement Learning, DeepSeek-AI et al., 2025
>
> ---
> **Q5:** I am missing some intuition on why RL is causing CoT length to go down (Figure 2c). What is happening to the model rollouts? Is it learning to merge steps, or do less backtracking, etc?
>
> **A5:** Thank you for this insightful question. Our analysis reveals that RL is teaching the model to **compress heterogeneous reasoning steps** by eliminating exploratory paths and directly reaching optimal solutions. We illustrate this with a concrete example from our evaluation dataset.
>
> **Example: Double Modular Exponentiation Problem**
>
> Task: Compute `((39³ % 10)^1000) % 1000` and compare with target 17.
>
> **Longer CoT (1965 chars) - Pre-RL behavior:**
>
> ```
> <think>
> First, calculate 39³ mod 10:
>   39 ≡ 9 mod 10
>   39² ≡ 9² ≡ 81 ≡ 1 mod 10
>   39³ ≡ 9 × 1 ≡ 9 mod 10
>
> Next, calculate 9^1000 mod 1000.
> We can use properties of modular arithmetic:
>   9² = 81
>   9⁴ = 81² = 6561 ≡ 561 mod 1000
>   9⁸ = 561² = 314721 ≡ 721 mod 1000
>   9¹⁶ = 721² = 519841 ≡ 841 mod 1000
>   ... [continues through 9^512]
>
> Instead of calculating this large product directly,
> we can use Python to compute it efficiently:
>   pow(9, 1000, 1000) = 1
>
> Since 1 ≠ 17, return []
> </think>
>
> ```
>
> **Shorter CoT (892 chars) - Post-RL behavior:**
>
> ```
> <think>
> Step 1: Calculate 39³ % 10 = 9
>
> Step 2: Calculate 9^1000 % 1000
>   Using pow(9, 1000, 1000) = 401
>
> Step 3: Compare 401 ≠ 17
>   Therefore, return []
> </think>
>
> ```
>
> **What RL is Learning:**
>
> 1. **Merging steps:** The model consolidates the exploratory phase (manual binary exponentiation attempts) with the execution phase (calling pow function) into a direct tool invocation.
> 2. **Eliminating backtracking:** The longer response exhibits a "try-realize-pivot" pattern (attempt manual calculation → recognize complexity → switch to programmatic approach). RL removes this exploratory detour.
> 3. **Preserving correctness:** Both responses reach the correct answer, but the shorter one achieves it through a more direct reasoning path that skips intermediate exploratory work.
>
> This compression reflects the model learning to identify and execute the most efficient solution strategy without sacrificing accuracy—a natural outcome when RL optimizes for both task success and efficiency.
>
> ---
> **Q6:** Similar question for Figure 4: what happens as the model goes from CoT length 24->5 - is it combining lots of steps together?
>
> **A6.** In short, the model is effectively *combining many steps into one*. When using only 5 CoT steps, each step typically solves multiple operators at once (e.g., computing 1+2+3+4=10 within a single step). In contrast, under a 24-step decomposition, each step handles only a single operator (e.g., 1+2). The reduction from 24 to 5 steps is thus driven by the model’s ability to solve more complex sub-tasks per step. This “step fusion” behavior is exactly what allows the model to reduce the total number of reasoning steps.

---

> ### Author Response · Authors · 2025-11-21
>
> **Q7:** For task difficulty 24, seems to only get better with more CoT length. does this level off?
>
> **A7.** Yes, it does level off. For difficulty-24 tasks, the model has only been exposed to CoTs up to length 24 during training. Given its limited capability, using all 24 steps appears optimal, mirroring the behavior of many real-world models that have not been trained on longer CoTs and have weak single-step problem-solving ability, making “longer is better” seem true. Once we increase task difficulty (e.g., difficulty 32) and allow the model to use longer CoTs, the weakness of excessively long chains becomes clear: as shown by the orange curve, accuracy at length 32 is lower than at length 16. This demonstrates that the “monotonic improvement” at difficulty 24 is an artifact of training-range and capability limits, not a fundamental property of CoT.
>
> ---
>
> Thank you again for the insightful comments. If you find it satisfactory, we respectfully hope that you can re-evaluate our work. We are happy to address your further concerns.

---

### Official Review · Reviewer_K7Ei · 2025-11-01

**Soundness:** 4
**Presentation:** 3
**Contribution:** 3
**Rating:** 8
**Confidence:** 4

**Summary:**

This paper reveals that the performance of Chain-of-Thought (CoT) reasoning does not uniformly improve with length; rather, accuracy follows an inverted U-shaped curve. The authors demonstrate that an optimal CoT length exists, which is dependent on several factors:
- Task Difficulty: More difficult problems require longer optimal CoT lengths to solve effectively.
- Model Capability: The optimal length varies inversely with model capability. More capable models tend to favor shorter, more efficient reasoning paths.
- Training Dynamics: Reinforcement Learning (RL) training can also induce this simplicity bias, guiding models to gravitate towards shorter, more optimal CoT lengths as their accuracy improves.
Based on these insights, the authors propose a theoretical framework that models the trade-off between task decomposition and error accumulation. They demonstrate that this insight can be practically applied to achieve better accuracy by aligning models with this optimal length during both training and inference.

**Strengths:**

1. The paper is well-written, clearly organized, and easy to follow. The authors articulate their core argument effectively.
2. The core finding is "an optimal Chain-of-Thought (CoT) length exists". This conclusion is convincingly demonstrated. The results of most experiments are very clear.
3. This study not only analyzes the results, but also provides guidance for the reasoning process of practical models, which has also shown positive effects in experiments.

**Weaknesses:**

The controlled experiments described in Section 3 may be problematic because factors other than CoT length were altered. For instance, the long and short CoT solutions differ not only in total length but also in how they approach problem-solving: short CoTs take fewer but longer steps, whereas long CoTs take more but shorter steps. Since the paper measures CoT length by the number of steps and controls this variable in the experiments, variations in step length and the more complex operations used in shorter CoTs could act as confounding factors, weakening the validity of the claimed causal link between length and performance.

**Questions:**

1. The sampling of model sizes in Figure 2(a) is sparse, making the "simplicity bias" trend less conclusive (e.g., the 72B model's optimal length is slightly longer than the 32B's). Adding intermediate models, like the 14B, would help create a more convincing trend line. Additionally, adding the MOE model would be better.

2. In Section 2, regarding the impact of RL on CoT length, an additional set of experiments using models of other scales should be included. The current results only demonstrate that RL can reduce CoT length. (While other studies have concluded that RL can increase CoT length, your intended proof here is not merely about whether CoT length increases or decreases, but rather that it converges to the optimal length—this requires verification through different trends in CoT length under different conditions.)

3. In the current experiments, most only examine the model's performance at a specific CoT length by manually controlling the length (see Fig. 4). However, I am more curious about the difference between the CoT length independently chosen by the RL-trained model during inference (both before and after RL) and the optimal CoT length. Can it be experimentally demonstrated that sufficient RL training enables the CoT to converge to the true optimal length?

4. Inconsistency in Figure 2: The caption for Figure 2(b) states it is "with the 7B model," whereas the main text (line 152) references the "Qwen1.5B-Instruct model" when discussing this result.

5. The definitions for symbols in the theoretical model could be strengthened. For instance, the constant 'C' introduced in Equation (1) (and later in Theorem 4.3) is defined somewhat informally.

6. Recent related work about theoretical understanding of CoT: "How Likely Do LLMs with CoT Mimic Human Reasoning?" and "Correlation or Causation: Analyzing the Causal Structures of LLM and LRM Reasoning Process".

---

> ### Author Response · Authors · 2025-11-21
>
> We thank the reviewer for the positive and encouraging assessment of our paper’s clarity, organization, and core contributions. We appreciate the recognition that our evidence for an optimal CoT length is convincing, and that our analysis offers actionable guidance for practical reasoning models with demonstrated empirical benefits. We address your concerns below.
>
> ---
>
> **W:** The long and short CoT solutions differ not only in total length but also in how they approach problem-solving: short CoTs take fewer but longer steps, whereas long CoTs take more but shorter steps.
>
> **A:** Thank you for the observation. We fully agree that short and long CoTs differ not only in total length but also in the granularity of their reasoning steps. In our framework, we explicitly define CoT length as the *number of decomposed subtasks*, rather than assuming that steps across different lengths are identical. Consequently, shorter CoTs naturally contain fewer but more complex steps, each covering more operations or tokens. This increased per-step difficulty is an intended part of our task-decomposition formulation and is precisely what our analysis captures.
>
> ---
>
> **Q1:** The sampling of model sizes in Figure 2(a) is sparse, making the "simplicity bias" trend less conclusive (e.g., the 72B model's optimal length is slightly longer than the 32B's). Adding intermediate models, like the 14B, would help create a more convincing trend line. Additionally, adding the MOE model would be better.
>
> **A1:** Thank you for the insightful suggestion. We agree that denser sampling of model scales can strengthen the visibility of the trend.
>
> - **On including 14B model:** Following your suggestion, we additionally evaluated the **14B model** (76.4 accuracy) under the same experimental settings. The results confirm our hypothesis: **the 14B model’s optimal CoT length is 7**, which lies cleanly between the 7B with optimal length 10 and 32B model with optimal length 2, thereby reinforcing the monotonic trend with respect to model capability. We have updated Figure 2(a) with 14B model.
> - **On including MoE models:** We appreciate the reviewer’s interest in whether MoE models can be incorporated into our scaling-law analysis. In practice, it is extremely challenging to find an open-source MoE *family* that offers multiple model sizes suitable for controlled scaling comparisons. For instance, Qwen only releases **Qwen2-57B-A14B**, Mistral provides **Mixtral 8×7B**, and DeepSeek only publishes **DeepSeek-MoE-16B-Chat**—each essentially a single checkpoint rather than a scalable series. As a result, evaluating CoT length scaling *within* an MoE family is currently infeasible.
>
>     A natural question is whether MoE models can be merged into dense-model scaling laws. Our findings suggest the answer is **no**. Using Qwen2-57B-A14B as an illustrative example (the MoE model closest to the Qwen2.5 dense family), we observe that despite its large parameter count, it exhibits **notably weaker reasoning performance**: on the MATH dataset, it achieves **43.0%**, compared to **49.8%** for the much smaller dense **Qwen2.5-7B**. This makes it difficult to position MoE models along a coherent “model capability axis,” as their size does not reliably translate into reasoning ability. This observation aligns with recent findings in the literature. Jelassi et al. [1] show both theoretically and empirically that MoE architectures are **more effective at memorization than reasoning**, largely because MoEs tend to memorize training patterns rather than acquire transferable reasoning skills.
>
>     Taken together, both our empirical results and prior research indicate that MoE models behave qualitatively differently from dense models in reasoning settings. Consequently, it is neither appropriate nor meaningful to include MoEs in our CoT-length scaling analysis at this stage.
>
>
> **Reference:**
>
> [1] Mixture of Parrots: Experts improve memorization more than reasoning, Jelassi et al., 2025

---

> ### Author Response · Authors · 2025-11-21
>
> **Q3:** In the current experiments, most only examine the model's performance at a specific CoT length by manually controlling the length (see Fig. 4). However, I am more curious about the difference between the CoT length independently chosen by the RL-trained model during inference (both before and after RL) and the optimal CoT length. Can it be experimentally demonstrated that sufficient RL training enables the CoT to converge to the true optimal length?
>
> **A3.** **Before addressing Q2, let me first clarify Q3**, as the experiment underlying Q3 is directly helpful for interpreting Q2. The experiments **do not manually force a fixed CoT length** at inference time. In the synthetic setup, each question comes with demonstrations of multiple CoT lengths, so the model naturally develops its own length preference. Figure 4 tracks how this preference shifts under RL, as reflected by **the probability (shown on the y-axis) that the model *spontaneously* selects CoTs of different lengths**. The base model spreads probability across lengths around 5, 12, and 24, but as RL training progresses, this distribution shifts to shorter CoTs and finally collapses toward the length 5, which is the accuracy-optimal length under controlled evaluation. This provides direct evidence that, with sufficient RL training, the model’s self-selected CoT length converges to the optimal length rather than simply becoming shorter or longer. We have revised Sec 3.4 IV to make this more clear.
>
> ---
>
> **Q2:** In Section 2, regarding the impact of RL on CoT length, an additional set of experiments using models of other scales should be included. The current results only demonstrate that RL can reduce CoT length.
>
> **A2:** We agree that the key question is not whether RL increases or decreases CoT length per se, but whether it calibrates the length toward the task-dependent optimum. Our real-world experiments already show that RL tends to shorten CoT length, while prior work [1,2] has reported cases where RL *increases* CoT length. Taken together, these observations suggest that RL is adjusting CoT length rather than monotonically pushing it up or down.
>
> To verify this more rigorously, in Sec. 3.2 (IV), Fig. 4 we train a synthetic model with explicit control over CoT length. Before RL, the model roughly spreads its probability mass over multiple lengths. **After RL, its behavior collapses to almost always selecting the CoT length that achieves the highest accuracy under each condition**. In addition, Corollary 4.4 provides a theoretical understanding that RL will automatically prefer the optimal length and hence calibrates the CoT length. Therefore, the main purpose of our real-world experiments is to verify the existence and scaling behaviour of the optimal CoT length.
>
> **Reference:**
>
> [1] Light-R1: Curriculum SFT, DPO and RL for Long COT from Scratch and Beyond, Wen et al, 2025
>
> [2] DeepSeek-R1: Incentivizing Reasoning Capability in LLMs via Reinforcement Learning, DeepSeek-AI et al., 2025
>
> ---
>
> **Q4:** Inconsistency in Figure 2: The caption for Figure 2(b) states it is "with the 7B model," whereas the main text (line 152) references the "Qwen1.5B-Instruct model" when discussing this result.
>
> **A4.** Thank you for catching this. You are absolutely right: the caption and the main text referenced different model sizes. This was an editing oversight. We have corrected the inconsistency and ensured that both the caption and the discussion now **refer to the 7B model**. The appendix includes results for the 1.5B model and other sizes as well. We appreciate you for pointing this out.
>
> ---
>
> **Q5:** The definitions for symbols in the theoretical model could be strengthened. For instance, the constant 'C' introduced in Equation (1) (and later in Theorem 4.3) is defined somewhat informally.
>
> **A5:** We agree that the definition of the constant *C* should be stated more rigorously. We have restore a complete set of formal definitions for **C**, **T**, and **M** in the revision. Below, we provide the exact definitions consistent with our arithmetic-task setup:
>
> - **Task difficulty $T$.**
>
>     Let $T$ denote the total number of operators in the given arithmetic problem, which serves as a problem-dependent proxy for task difficulty.
>
> - **Model capability $M$**.
>
>     For each model with parameters $\theta$, we define its capability M($\theta$) using the *reasoning boundary*. Intuitively, $M(\theta)$ represents the largest subproblem size the model can reliably solve in a single reasoning step.
>
> - **Training-difficulty constant **$C$**.**
>
>     The constant $C$ denotes the maximum task difficulty the model family is trained to handle, which is the largest operator count that appears in the training dataset.
>
> Therefore, $T$, $M$, and $C$ all refer to operator counts, and it is natural to analyze the simple linear case, where error rate is the ratio of subtask operators and maximum capable operators.

---

> ### Author Response · Authors · 2025-11-21
>
> **Q6:** Recent related work about theoretical understanding of CoT: "How Likely Do LLMs with CoT Mimic Human Reasoning?" and "Correlation or Causation: Analyzing the Causal Structures of LLM and LRM Reasoning Process".
>
> **A6.** Thank you for highlighting these two relevant studies. Bao et al.[1] investigates the causal structures underlying CoT generation and shows that LLMs often mix genuine reasoning with post-hoc justification, with different SCM types predicting faithfulness and consistency behaviors. Fu et al.[2] examines how different training paradigms—such as SFT, distillation, and RLVR—shape the causal robustness of model reasoning, emphasizing that methods like RLVR can strengthen causal alignment while others may inadvertently amplify spurious features.
>
> Our work differs in both scope and objective. These studies focus on *identifying and characterizing* the latent causal structures behind model reasoning, whereas our work develops a *predictive and quantitative framework* centered on how CoT length interacts with task difficulty, model capability, and error accumulation. Rather than analyzing causal faithfulness of reasoning traces, we aim to explain why an optimal CoT length exists, how it scales, and how RL converges to it.
>
> We appreciate the reviewer’s suggestion, and **we have incorporated both papers into our Related Work section for a clearer contextualization of their contributions relative to ours.**
>
> [1] How Likely Do LLMs with CoT Mimic Human Reasoning?, Bao et al., 2024
>
> [2] Correlation or Causation: Analyzing the Causal Structures of LLM and LRM Reasoning Process, Fu et al., 2025
>
> ---
>
> Thank you again for the insightful comments. If you find it satisfactory, we respectfully hope that you can re-evaluate our work. We are happy to address your further concerns.

---

> > ### Comment · Reviewer_K7Ei · 2025-11-26
> >
> > Thanks for the detailed response, which resolves my concerns. I'll keep my current rating and look forward to the reviewers' discussion.

---

### Official Review · Reviewer_Bgbw · 2025-11-02

**Soundness:** 3
**Presentation:** 3
**Contribution:** 2
**Rating:** 2
**Confidence:** 4

**Summary:**

This paper presents several insights about the length of reasoning chains within long chain-of-thought reasoning.

**Strengths:**

1. Overall the paper is pretty well written and I was able to follow the main points.
2. The insight regarding step-wise computation also increasing for difficult instances is an interesting one that as far as I know as not covered extensively in previous papers.

**Weaknesses:**

The paper is mainly insight-driven, but as far as I can tell many of the insights presented in the paper have already been uncovered by previous work that was not cited:

1. Insights related to long CoTs not being better are covered by Jiang et al., which was not cited.
2. The idea of error accumulation in CoTs being responsible for long chains of thought being less successful was published in Schaeffer et al. 2023.
3. Adaptive length-filtered voting was examined by Fu et al., which was cited in the paper but not cited in the appropriate section.

References:
- What Makes a Good Reasoning Chain? Uncovering Structural Patterns in Long Chain-of-Thought Reasoning. Jiang et al. EMNLP 2025.
- Are Emergent Abilities of Large Language Models a Mirage? Schaeffer et al. NeurIPS 2023.
- Complexity-Based Prompting for Multi-Step Reasoning. Fu et al. ICLR 2023.

**Questions:**

1. If you were to refine the main claims of the paper based on what has already presented in the previous work that I cited, what do you think are the remaining most valuable insights?
2. I'm not sure if the term "scaling law" is appropriate in this case. This is a bit of a nuance, but usually when we talk about "large scale" it's about things like data, compute, etc. and not some 10s of thousands of tokens in a reasoning trace. Maybe you could consider a different term here?

---

> ### Author Response · Authors · 2025-11-21
>
> Dear Reviewer Bgbw,
>
> **We sincerely appreciate your thoughtful feedback.** After reviewing your comments, we believe that several misunderstandings about the relationships between our work and prior literature may have unintentionally led to an underestimation of our contributions. We clarify these points below and hope this resolves the concerns.
>
> ---
>
> **Q1. Novelty compared to related literature**
>
> **A1.** Thanks for bringing up these related works. Below, let us elaborate their relationships point by point.
>
> **(1.1) Relation to Jiang et al. (EMNLP 2025)**
>
> Thank you for raising this connection. We now cite Jiang et al. explicitly and have revised the related-work section to elaborate on the relationship clearly.
>
> To clarify: our work is the first in the literature to reveal and theoretically justify that CoT accuracy follows an inverted-U curve and that an optimal CoT length exists. An earlier draft of our work was publicly available and Jiang et al. explicitly cited this early draft and built upon our insight that longer CoTs are not always better, providing a more fine-grained empirical characterization of structural patterns in long chains.
>
> Comparing the two works, Jiang et al not explain *why* such patterns arise, while ours contributes:
>
> - the **first principled theoretical framework** explaining the non-monotonic dependence of accuracy on CoT length,
> - the **first identification and theoretical justification** of the inverted-U curve and the optimal length \(N^*\),
> - a unified account of reasoning depth, model capability, and task difficulty.
>
> Jiang et al.’s empirical observations are consistent with our findings, but our work formulates, explains, and theoretically grounds the phenomenon itself in the first place. Therefore, we think that our work does provide unique values to the literature.
>
> **(1.2) Relation to Schaeffer et al. (NeurIPS 2023)**
>
> We appreciate the reviewer pointing out this connection. We have added explicit citations and a dedicated discussion in the revision.
>
> Schaeffer et al. use error accumulation to explain emergent abilities via a monotonic \(p^L\) formulation with fixed reasoning length, based on which they show that i**mproved token-level accuracy yields better end-to-end accuracy.** Instead, our work is the first to leverage error accumulation to analyze the influence of CoT length and derive the existence of **an optimal CoT length** and **non-monotonic performance ( inverted-U accuracy curve).** This provides valuable insights of the reasoning behaviors of LLMs and how to design efficient CoTs.
>
> We have now cited and elaborated this relationship in Section 6.
>
> **(1.3) Relation to Fu et al. (ICLR 2023)**
>
> We thank the reviewer for highlighting this. We now cite Fu et al. in the section 5.2 (Adaptive Length-filtered Vote) and elaborate on the distinction in the revision.
>
> Fu et al. propose filtering out *short* CoTs, which worked for smaller 2023-era models where “longer is better.” Our findings demonstrate that modern LLMs show **clear non-monotonicity**, and thus effective filtering should remove both **too-short** and **too-long** chains. This is directly motivated by our theoretical model, which estimates the optimal length range. This also provides valuable new insights into the literature.
>
> ---
>
> **Q2. Remaining valuable insights after accounting for prior work**
>
> **A2.** Even when considering all cited literature, our contributions remain novel and foundational:
>
> 1. **We are the first to reveal and theoretically justify the inverted-U relationship** between CoT length and accuracy, providing a principled explanation for overthinking in LLMs.
> 2. **We introduce a unified model that connects reasoning depth, model capability, and task difficulty**—far beyond token-level formulations.
> 3. We provide the first **filtering method** for estimating **an optimal CoT length range** and show its empirical robustness. This method is built on the theory by filtering **both overly short and overly long chains**, improving performance across models and tasks.
> 4. Our work has inspired and driven a series of follow-up works exploring CoT compression and reasoning-length optimization (see [1-5] for a few examples).
>
> Therefore, after accounting these related work, we belive that our work still has its unique value adding to the literature.
>
> **References:**
>
> [1] Concise Reasoning via Reinforcement Learning, Fatemi et al., 2025
>
> [2] How Well do LLMs Compress Their Own Chain-of-Thought? A Token Complexity Approach, Lee et al., 2025
>
> [3] Between Underthinking and Overthinking: An Empirical Study of Reasoning Length and correctness in LLMs, Su et al., 2025
>
> [4] ShorterBetter: Guiding Reasoning Models to Find Optimal Inference Length for Efficient Reasoning, Yi et al., 2025
>
> [5] AdaCoT: Pareto-Optimal Adaptive Chain-of-Thought Triggering via Reinforcement Learning, Lou et al., 2025.

---

> ### Author Response · Authors · 2025-11-21
>
> **Q3.** I'm not sure if the term "scaling law" is appropriate in this case. This is a bit of a nuance, but usually when we talk about "large scale" it's about things like data, compute, etc. and not some 10s of thousands of tokens in a reasoning trace. Maybe you could consider a different term here?
>
> **A3.** Thank you for raising this nuance. Our use of the term *scaling law* follows the notion of *test-time scaling laws* introduced in OpenAI’s o1 model [1], where performance is shown to improve as test-time compute increases. This terminology has since been adopted by many follow-up works and have become a common terminology [2,3,4] (as discussed in Sec. 6) that analyze how allocating more inference-time computation affects accuracy.
>
> Besides, our discovered scaling behaviors also cover the scaling of CoT length w.r.t. other scaling dimensions, like model parameters (eg Fig 2a) and task complexity (eg Fig 2b), where the former is more well-known in “scaling law” studies. We believe that revealing these different scaling behaviors provides a more holistic understanding of the influencing factors of CoT length.
>
> Because of these considerations, we adopt the “scaling” terminology. But please let us know if there is a better candidate and we are happy to consider. Thanks!
>
> **Reference:**
>
> [1] *Learning to Reason with LLMs*, OpenAI, 2024.
>
> [2] Scaling llm test-time compute optimally can be more effective than scaling model parameters, Snell et al., 2024.
>
> [3] Scaling inference computation: Compute-optimal inference for problem-solving with language models. Wu et al., 2024.
>
> [4] Large Language Monkeys: Scaling Inference Compute with Repeated Sampling, Brown et al., 2024.
>
> ---
>
> **Final Remark.** We hope the above clarifications help fully address your concerns. As pointed out by Reviewers K7Ei, seMZ, and pd7o, this work “makes important contributions toward addressing the fundamental question of how long reasoning should be in LLMs, with significant implications for designing models under a reasoning budget.”
>
> Since concerns about novelty appear to be the main reason behind the initial rating, we would kindly appreciate a reevaluation if the clarifications above resolve those doubts. Please feel free to let us know if any additional explanation would be helpful. Thank you very much for your time and consideration.

---

### Meta-Review · Area_Chair_zCd3 · 2025-12-28

**Summary:**

This paper investigates the relationship between CoT reasoning length and performance, revealing an inverted U-shaped curve where accuracy initially rises but eventually declines due to error accumulation. The authors propose a theoretical framework demonstrating that optimal CoT length scales with task difficulty but decreases as model capability improves. Reviewers praised the work for its counter-intuitive empirical findings and rigorous synthetic experiments. While concerns were initially raised regarding novelty compared to concurrent works and the definition of task difficulty, the authors successfully differentiated their theoretical contribution and validated their findings with additional experiments on the 14B model and self-correction mechanisms. Given the strong theoretical grounding and practical insights for model efficiency, I recommend accepting this paper for ICLR 2026.

**Reviewer Concerns:**

Concerns addressed by the rebuttal:
- Experimental confounding factors and model sampling by reviewer K7Ei
- Theoretical clarifications and definitions by reviewer K7Ei and reviewer pd7o
- Impact of self-correction on CoT length by reviewer seMZ
- Generality of findings across tasks and models by reviewer pd7o

**Reviewer Scores:**

One reviewer with a rating of 8 explicitly mentioned that they would like to maintain their score. Other reviewers are likely to raise their rating based on the responses by the authors.

---

### Decision · Program_Chairs · 2026-01-26

Accept (Poster)